
# A new method (M³Fusion-v1) for combining observations and multiple model output for an improved estimate of the global surface ozone distribution

Kai-Lan Chang[1, 2], Owen R. Cooper[2, 3], J. Jason West[4], Marc L. Serre[4], Martin G. Schultz[5], Meiyun Lin[6, 7], Virginie Marécal[8], Béatrice Josse[8], Makoto Deushi[9], Kengo Sudo[10, 11], Junhua Liu[12, 13], and Christoph A. Keller[12, 13, 14]

[1]National Research Council Fellow
[2]NOAA Earth System Research Laboratory, Boulder, CO, USA
[3]Cooperative Institute for Research in Environmental Sciences, University of Colorado, Boulder, CO, USA
[4]Department of Environmental Sciences & Engineering, University of North Carolina, Chapel Hill, NC, USA
[5]Jülich Supercomputing Centre (JSC), Forschungszentrum Jülich, Jülich, Germany
[6]NOAA Geophysical Fluid Dynamics Laboratory, Princeton, NJ, USA
[7]Program in Atmospheric and Oceanic Sciences, Princeton University, Princeton, NJ, USA
[8]Météo-France, Centre National de Recherches Météorologiques, Toulouse, France
[9]Meteorological Research Institute (MRI), Tsukuba, Japan
[10]Graduate School of Environmental Studies, Nagoya University, Nagoya, Japan
[11]Japan Agency for Marine-Earth Science and Technology (JAMSTEC), Yokosuka, Japan
[12]NASA Goddard Space Flight Center, Greenbelt, MD, USA
[13]Universities Space Research Association, Columbia, MD, USA
[14]John A. Paulson School of Engineering and Applied Science, Harvard University, Cambridge, MA, USA

**Correspondence:** Kai-Lan Chang (kai-lan.chang@noaa.gov)

**Abstract.** We have developed a new statistical approach (M³Fusion) for combining surface ozone observations from thousands of monitoring sites around the world with the output from multiple atmospheric chemistry models to produce a global surface ozone distribution with greater accuracy than can be provided by any individual model. The ozone observations from 4766 monitoring sites were provided by the Tropospheric Ozone Assessment Report (TOAR) surface ozone database which

5     contains the world's largest collection of surface ozone metrics. Output from six models was provided by the participants of the Chemistry-Climate Model Initiative (CCMI) and NASA's Global Modeling and Assimilation Office (GMAO). We analyze the 6-month maximum of the maximum daily 8-hour average ozone value (DMA8) for relevance to ozone health impacts. We interpolate the irregularly-spaced observations onto a fine resolution grid by using integrated nested Laplace approximations, and compare the ozone field to each model in each world region. This method allows us to produce a global surface ozone

10    field based on TOAR observations, which we then use to select the combination of global models with the greatest skill in each of 8 world regions; models with greater skill in a particular region are given higher weight. This blended model product is bias-corrected within two degrees of observation locations to produce the final fused surface ozone product. We show that our fused product has an improved mean squared error compared to the simple multi-model ensemble mean.

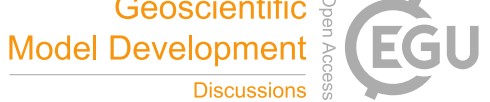

# 1 Introduction

Tropospheric ozone is a pollutant detrimental to human health and has been associated with a range of adverse cardiovascular and respiratory health effects due to short-term and long-term exposure (World Health Organization, 2005; Jerrett et al., 2009; US Environmental Protection Agency, 2013; Forouzanfar et al., 2015; Turner et al., 2016; Cohen et al., 2017). Assessing the

human health impacts of ozone on the global scale requires accurate exposure estimates at any given inhabited location (Shaddick et al., 2018). Due to the limited availability of surface ozone observations in many regions of the world (Fleming et al., 2018), global atmospheric chemistry models are required to calculate surface ozone exposure. Despite continual development and improvement, global models struggle in their ability to accurately simulate ozone in all regions of the world (Young et al., 2018). The ability to accurately simulate observed ozone at a particular location also varies between models, as demonstrated

by several multi-model comparisons (Stevenson et al., 2006; Young et al., 2013; Cooper et al., 2014).

A useful endeavor for producing an accurate representation of the global surface ozone distribution is to combine the output from many models in a way that takes advantage of the strengths of each model and minimizes the weaknesses. Such efforts have already been made for both climate and chemistry climate models. For example, multi-model output has been combined using a parametric approach, either by assigning an equal or optimum weight to each model (Stevenson et al., 2006; He and

Xiu, 2016; Braverman et al., 2017), or by tuning the initial conditions under different scenarios or parameterizations (Cariolle and Teyssèdre, 2007; Wu et al., 2008; Young et al., 2013). These approaches often assume that individual model biases will at least partly cancel by averaging or weighting, according to certain measures of predictive performance. Thus the combined model product is likely to be more accurate than a single model prediction, as has been shown for multi-model combinations of past or present day climate (Buser et al., 2009; Knutti et al., 2010; Weigel et al., 2010; Chandler, 2013).

For the case of simply averaging the output from multiple climate models, most studies either explicitly or implicitly assume that every model is independent and is a random sample from a distribution, with the true climate as its mean. This implies that the average of a set of models converges to the true climate as more and more models are added. This multi-model ensemble often outperforms any single model in terms of the predictive capability. Undeniably, when one has several dozen or hundreds of possible ensemble members, the most straightforward and efficient approach is to simply take the ensemble average, ignoring

the impact of potentially erroneous outlier ensemble members. Therefore, from a statistical point of view, one might argue that ruling out potentially erroneous ensemble members prior to conducting the ensemble mean would yield an even better result, especially if the overall number of ensemble members is small.

Combining model ensembles using a method more sophisticated than the simple average is a challenge because a meaningful model evaluation can rarely be condensed into a single metric, and there is no technique that can explicitly quantify the degree

of similarity (i.e both accuracy and precision) between two different spatial fields (Hyde et al., 2018). Indeed, Stainforth et al. (2007) concluded that any attempt to assign weights is, in principle, inappropriate. With a lack of appropriate criteria, the model weighting approach has not become a standard alternative to the ensemble average. Accordingly, there is presently no objective criterion for combining surface ozone estimates from a model ensemble to produce a surface ozone product with

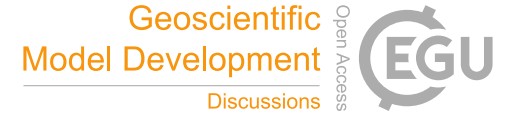



improved accuracy beyond that of any ensemble member or the simple ensemble mean. The absence of such a methodology is the motivation for this paper.

This paper presents a new statistical approach (M³Fusion) for combining surface ozone output from multiple atmospheric chemistry models with all available surface ozone observations to produce a global surface ozone distribution with far greater

accuracy than the multi-model ensemble mean. As described in greater detail below, this fused surface ozone product is constructed in three steps: 1) Ozone observations from all available surface ozone monitoring sites around the world are spatially interpolated to a smooth global field; 2) For each of 8 continental regions of the world 6 global atmospheric chemistry models are evaluated against the interpolated observed ozone field by a quadratic programming optimization, with the most accurate models receiving the highest weight; 3) the global ozone field derived from the polynomial equation is bias corrected within a

given distance from available observations. The final product is based on the annual maximum of the 6-month running mean of the maximum daily 8-hour average (DMA8), a metric that can be used to estimate human mortality due to long-term ozone exposure (Turner et al., 2016).

The fused surface ozone product will be provided to the Global Burden of Disease (GBD) team for their use in evaluating the global health burden of ambient ozone. Whereas the GBD studies used a sophisticated approach to estimate global $PM_{2.5}$

at high resolution, a much simpler approach was used for ozone. For $PM_{2.5}$, Brauer et al. (2012, 2015) and Shaddick et al. (2018) merged information from a single global model Tracer Model 5 (TM5), satellite estimates (Van Donkelaar et al., 2010), and available surface observations from networks. For ozone, however, only a simulation was used in recent GBD studies (Lim et al., 2012; Forouzanfar et al., 2015; Cohen et al., 2017). While the TM5 modeling group used observations to evaluate the model, no observations were used directly by the GBD team and model biases were not corrected. Therefore we expect our

work to significantly improve forthcoming GBD estimates, yielding a more accurate understanding of ozone exposure and its global health effects.

Section 2 provides details of the data sources and fusion process, including the techniques to register all data sources onto a common grid, and a statistical model to minimize the difference between interpolated observations and the multi-model combination. In Section 3 the results of employing these techniques are presented, including the mapping accuracy, evaluation

of regional model performance and the final multi-model bias correction. The paper concludes with a summary and discussion in Section 4.

## 2 Data and Method

### 2.1 Observations and model output

1. *Tropospheric Ozone Assessment Report (TOAR) database*: In this analysis, surface ozone observations are used to eval-

uate the performance of 6 global atmospheric chemistry models and to also bias-correct the multi-model surface ozone product. TOAR has produced the world's largest database of surface ozone metrics based on hourly observations at over 9000 sites around the globe (Schultz et al., 2017, available for download at: https://doi.org/10.1594/PANGAEA.876108). Spatial coverage is high in North America, Europe, South Korea and Japan, but much lower across the rest of the world





with very low data availability across Africa, the Middle East, Russia and India. In addition to data sparseness, other challenges, such as data inhomogeneity in time and the irregular spatial distribution of stations (Chang et al., 2017), make the comparison between model output and observations difficult without serious statistical modeling. While satellite retrievals have been utilized by previous works for quantifying the health impacts of $PM_{2.5}$ (Brauer et al., 2012, 2015), satellite retrievals of tropospheric ozone have limited sensitivity near the surface and are inadequate for this analysis (Gaudel et al., 2018).

TOAR has gathered ozone observations through 2014 at most sites, and has chosen 2008-2014 as a "present-day" window for more rigorous analysis. The purposes of the multi-year average are to reduce the effects of ozone interannual variability, which is largely driven by changes in meteorological conditions (Strode et al., 2015), and to increase the number of available sites than if we used a single year. In this analysis we focus on the annual maximum of the 6-month running mean of the maximum daily 8-hour average (DMA8) at every site in the TOAR database. This metric was selected because it aligns with the ozone metric used by Turner et al. (2016) to quantify the impact of long-term ozone exposure on human mortality. Hereinafter this quantity is simply referred to as "the ozone metric".

2. *Atmospheric chemistry model simulations*: We use output from models from phase 1 of the Chemistry-Climate Model Initiative (CCMI), downloaded from the Centre for Environmental Data Analysis (CEDA) database (http://archive.ceda.ac.uk). We choose four models (CHASER, GEOSCCM, MOCAGE and MRI-ESM1r1) that report hourly ozone output (Table 1). These particular simulations were part of CCMI's REF-C2 experiment (Morgenstern et al., 2017) which follows the WMO (2011) A1 scenario for ozone depleting substances, and RCP 6.0 for tropospheric ozone precursors, and aerosol and aerosol precursor emissions (Morgenstern et al., 2010) for the period 1960-2100. Even though the most appropriate experiment would have been the REF-C1SD, which aims for reproducing the past against observations, we use output from the REF-C2 simulation in this study, as the last year of the REF-C1SD was 2010, and would therefore not cover the most recent period where observations are available. However, the REF-C1SD simulation by the NOAA Geophysical Fluid Dynamics Laboratory (GFDL) AM3 model was available for the entire study period and was therefore selected for this analysis. In addition, we obtained output from the GEOS-5 nature run with chemistry (G5NR-Chem), provided by the NASA Global Modeling and Assimilation Office (GMAO), which we included in our analysis because of the model's very fine horizontal resolution (Hu et al., 2018), but the output was only available for July 2013 to June 2014.

The output from each individual model is shown in supplemental Fig S-1. Note that NASA G5NR-Chem has the finest resolution of these models, accordingly we aim to produce our final product on the same $0.125° \times 0.125°$ grid. However, even at this resolution the output is not street-resolving and thus will not capture urban scale variability in the regions with the highest population density.

In order to compare model output to observations, we need to register model output and observations to a common grid. This registration enables us to quantify the differences between the models and observations. Previous attempts have usually relied on a variant from a general statistical interpolation framework to combine incompatible spatial data (Gotway and Young, 2002;



Fuentes and Raftery, 2005; Gelfand and Sahu, 2010; Berrocal et al., 2012; Nguyen et al., 2012). Due to the highly irregular locations of ozone monitors around the globe, we use a kriging technique to build a statistical model, interpolate the ozone distribution based on the surrogate, and then project the global surface onto a common grid.

## 2.2 Fusion of observations and models

Following is a description of our method for fusing observations and output from multiple global atmospheric chemistry models to produce a surface ozone product with maximized accuracy. This method is known as Measurement and Multi-Model Fusion (version 1), or M$^3$Fusion (v1), and the code accompanies this manuscript on its associated Geoscientific Model Development discussion webpage. We consider a general framework of uncertainty quantification consisting of the following components (Kennedy and O'Hagan, 2001; Higdon et al., 2004; Chang and Guillas, 2018):

Observation = Reality + Random Error;

Reality = Model + Structured Bias,

Since this equation requires matching components (observations and model output) on a common grid, we use the interpolated observations to estimate an optimized weight for each model by a $L^2$ norm (details are given later), which means that we expect the multi-model combination to capture the general pattern of the surface ozone distribution in terms of their joint predictive

capability; and the model bias is considered as a model correction term. The difference between observation error and model bias is that the former term is assumed to be a normal noise with zero mean and constant variance; and the latter term is considered as a systematic and structured discrepancy (Williamson et al., 2015), which will be revealed as a spatial cluster across a poorly simulated region.

Due to this study's human health focus we do not consider ozone above the data-sparse oceans. Above land, large obser-

vational gaps are present across Africa, the Middle East, South America, and South and Southeast Asia, where the spatial interpolation is generally too uncertain to yield a reliable surface ozone approximation. The ozone estimates in these regions must come from either models or distant observations, neither of which is ideal. As a compromise strategy we fill these gaps with a weighted model product evaluated by the interpolated ozone observations. We propose the following procedure to combine model output and observations for data integration:

1. *Interpolating irregularly located monitoring observations to the model output grid*: We use a statistical kriging technique to interpolate ozone observations from irregularly located monitoring sites onto a $0.125° \times 0.125°$ grid, using the SPDE (stochastic partial differential equation) approach (Bolin and Lindgren, 2011; Lindgren et al., 2011). To differentiate this result from the actual observations in the TOAR database, we refer to this interpolated surface as the "spatially interpolated ozone".

We use a bilinear interpolation to smooth model output from coarser resolution to a $0.125° \times 0.125°$ grid (Jun et al., 2008). The ozone metric for each model was calculated for each single grid in each year, then averaged over 2008-2014 (except for NASA G5NR-Chem, which was already in fine resolution, but only available for 1 year).





2. *Weighting model output against spatially interpolated ozone by region*: We divide the global land surface into 8 regions (see Fig 2), roughly matching the continental outlines or major population regions. We adopt this regional approach because global models vary in their ability to simulate ozone in different regions of the world. Next we regress the observations on multi-model output by a constrained least square approach within each of the eight regions. Let $s_g$ be the grid cell at resolution $0.125° \times 0.125°$, $\hat{y}(s_g)$ be the interpolated observations, and $\{\eta_k(s_g); k = 1, \ldots, 6\}$ be the model output registered onto the same grid from the six models considered in this paper (table 1). The optimization equation is based on a constrained least squares approach:

$$\underset{\{\alpha_r, \beta_{rk}; k=1,\ldots,6\}}{\text{minimize}} \sum_{s_g \in \text{Region } r} \left( \hat{y}(s_g) - \alpha_r - \sum_{k=1}^{6} \beta_{rk} \eta_k(s_g) \right)^2, \tag{1}$$

subject to $\sum_{k=1}^{6} \beta_{rk} = 1$ and $\beta_{rk} \geq 0$.

where $\alpha_r$ is a constant that allows adjustment to the overall (regional) underestimation or overestimation, $\beta_{rk}$ is an optimal weight for the $k$-th model in region $r$. Note that we constrain the weights to be positive and sum to 1 in order to achieve a better physical interpretability. A constant offset $\alpha_r$ is included to guarantee that the residuals from this optimization have a zero mean. We refer to this intermittent model weighting product as the "multi-model composite".

3. *Correcting multi-model bias*: A common practice of studying the model discrepancy in the spatial fields is to fit a statistical model for their differences from observations on the whole spatial domain, to see whether or not these residuals reveal any structured spatial pattern (Jun and Stein, 2004; Sang et al., 2011). However, in our case the regular grid observation field is obtained from spatial kriging, such that in many data sparse regions we don't actually have observed ozone, which prevents us from correcting the model in these regions. Instead, we correct the model bias based on the distance to the nearest monitoring station, but we ignore the differences between the multi-model composite and the interpolated observations in the sparsely monitored regions. In our approach we only correct the output grid where there is at least one observational station within a 2 degree radial distance of the grid cell in question (i.e. the distance to the nearest station is less than $2°$). We then end up using

$$\begin{cases} \hat{y}(s_g), & \text{if } s_g \text{ is within a 2 degree radial distance of the station;} \\ \alpha_r + \sum_{k=1}^{6} \beta_{rk} \eta_k(s_g), & \text{otherwise,} \end{cases}$$

to generate our high resolution global surface ozone estimate. We refer to the final outcome as the "fused surface ozone product".

## 2.3 Implementation details

All computations in our methodology are implemented in R. The matching of interpolated observations and gridded model output does not involve complex statistical modeling. The remaining details of the methodology are as follows:



- *Kriging using the INLA-SPDE approach* (step 1): Kriging is a procedure used to statistically interpolate irregularly spaced and/or sparse observed data onto a regular and dense grid, based on a weighted average of the fitted surrogate model in the neighborhood of the grid. We assume that the global ozone distribution can be approximated by a Gaussian spatial process (GP) with a constant mean and Matérn covariance function (Stein, 2012). The GP fitting typically involves

a cubic complexity, and thus is computationally expensive for large spatial data sets. Therefore several alternatives have been developed to address the large $n$ issue by using a reduced set of data (Cressie and Johannesson, 2008; Banerjee et al., 2012), tapering the covariance between two grid point to zero if their distance is beyond a certain range (Furrer and Sain, 2009; Sang and Huang, 2012), and/or evaluating the covariance only through the specification of a neighborhood system (also known as the Gaussian Markov random field) (Rue et al., 2009; Lindgren et al., 2011).

In this study we carry out the spatial interpolation by using the combination of the *integrated nested Laplacian approximations* (INLA) framework (Rue et al., 2009), and the *stochastic partial differential equation* (SPDE) technique (Lindgren et al., 2011), available as an R package (http://www.r-inla.org/) (Lindgren and Rue, 2015). The details of this technique are rather complex and the reader is referred to the original paper (Lindgren et al., 2011), however we describe the key component of this INLA-SPDE technique in the Appendix. INLA-SPDE spatial modeling has proven

to be effective in a wide range of applications (Cameletti et al., 2013; Shaddick and Zidek, 2015; Heath et al., 2016; Liu and Guillas, 2017; Rue et al., 2017). We chose this technique because it manages a fairly large and complex spatial field in a relatively efficient way (Rue and Held, 2005), and allows an extension for nonstationarity on the sphere (Bolin and Lindgren, 2011; Chang et al., 2015). Notably, a recent study elaborately compared dozens of spatial modeling approaches, and the results suggest that almost all of these approaches can achieve a similar performance in terms of their

predictive accuracy, albeit with very different computation times (Heaton et al., 2018). Therefore, we expect that the choice of spatial modeling approach is not the most crucial component in our data fusion process as long as the analysis is carried out in a rigorous way (i.e. through the statistical model selection and diagnostics).

We carry out the statistical interpolation via the following steps: (1) calculate the ozone metric at each TOAR site and for every year in 2008-2014; (2) perform the statistical interpolation using all available sites with their exact coordinates,

and project the surface onto a $0.125° \times 0.125°$ spherical grid for every year; (3) average these surfaces over the 7 years to yield an observation-based present-day ozone distribution. We expect that this aggregation will smooth out at least some of the potential uncertainties. The kriging can be seen as a nonparametric regression problem, therefore a statistical assessment of fitted quality must be considered to select the best representation to the data (Hoeting et al., 2006). Further details on the statistical model selection procedure are provided in Appendix A.

- *Quadratic programming for multiple spatial fields* (step 2): The success of our method depends on that a model should not only be evaluated by its overall mean error (such as RMSE or mean absolute error), but also by its spatial structure (e.g. placement or arrangement of the curvature and domain shape). For example, consider the synthetic one dimensional spatial curvatures in Fig. 1, with the dark solid curve indicating the true process, and the dashed and dotted lines representing the output from two different models, A and B. Even though model A was closer to the true process in terms





of lower mean error than model B, our optimization algorithm gives more weight to model B because it reproduces the correct curvature. To avoid altering the underlying fixed spatial structure of each model's surface ozone distribution, our approach is designed to give greater weight to a model with a closer spatial pattern to the interpolated observations (regardless of the RMSE). Such a model can then be scaled to achieve a better fit to the interpolated observations (e.g. see later results for CHASER).

Due to the sparsity of stations exhibited in many regions, we use a pre-defined geometric boundary to differentiate regions. A more meaningful physical boundary (i.e., regions with similar chemical regimes, or major features such as deserts, mountain ranges or water bodies) might be determined using a cluster analysis technique (Hyde et al., 2018), but such a step is beyond the scope of this paper.

Since we partition the global land surface into eight regions and evaluate the models individually, inevitably there will be disjointed boundaries between regions. The boundaries between North and South America, or between East Asia and Oceania, fall mostly in the oceans, so we do not need to adjust these regions. However, we should make an adjustment to disjointed boundaries that fall across inhabited areas (see supplementary Fig S-2 for the illustration). As an example of our method, consider the boundary between East Asia and Russia near 50°N. We increase the northern boundary of East Asia to 55°N and decrease the southern boundary of Russia to 45°N, to create an overlapping intersection, and then fit cubic splines (performed for each grid cell) with knots placed at every 2 degree grid cell (Wood et al., 2008; Bakka et al., 2018). The result is a smoothed transition between the two regions.

- *Identifying structural biases in multi-model composite* (step 3): Even though the multi-model composite is a product evaluated from a regional optimization procedure, it still cannot account for all of the real-world ozone variability, thus we apply a final correction by comparing the multi-model composite to the interpolated observations. Instead of treating the differences in each grid cell directly as a correction term, a common approach uses a statistical model to retrieve the major mode of variability and capture the bias pattern from the residual. With a statistical approach, we can expect that this approximation can remove additional noise (i.e. small fluctuations will be filtered in order to keep the general spatial pattern produced by the models).

  If the model adequately simulates the ozone distribution (up to a level shift and a scale factor), then there is no relevant information in these residuals. On the other hand, if the model does not properly represent the local structure, then the residuals should exhibit a signal of the discrepancy in that region (Guillas et al., 2006; Williamson et al., 2015). Nevertheless, the highly irregularly distributed station network makes this spatial modeling approach undesirable, because the interpolated ozone at an unobserved location in a sparse region is unreliable for model correction. Thus, for each model grid cell, we apply a bias correction only if the distance to the nearest station is less than 2 degrees. The global coverage over land of this correction range is 14.4%. Since a large fraction of the model grid cells will not be corrected, pursuing another statistical modeling method to retrieve the major mode of variability becomes unnecessary; instead, we use the differences from interpolated observations directly as correction terms. We provide maps of the direct residual and the bias estimated from the statistical model in the supplemental figure S-5.




## 3  Results

### 3.1  Mapping and uncertainty

Ground based measurements were available from 4766 stations reported in the TOAR database (Schultz et al., 2017). To illustrate the spatial coverage of the database, Fig 2 shows the ozone metric discretized to a $2° \times 2°$ grid (a finer resolution

will be too obscure for illustrative purposes), averaged over the period 2008-2014. This figure also shows our regionalized classification, including Africa, North America, South America, East Asia, Southeast and central Asia, Europe, Oceania, and Russia. Note that dense station networks are found in North America, Europe and East Asia (mostly in Japan and South Korea), while monitoring sites are more widely scattered across the remaining regions. The highest average ozone levels (> 70 ppb) are found at sites in China, South Korea, Japan, Taiwan, India, Greece, California and Mexico City.

Fig 3(a) shows the spatially interpolated surface in each cell. For each grid cell, there is an underlying (posterior) probability distribution which incorporates information about the interpolation uncertainty. Fig. 3(b) shows the half-width of the 95% posterior credible interval in each cell (Shaddick et al., 2018). From the spatial pattern of uncertainty, we can see that relatively higher uncertainties are expected in Africa, the Middle East, South Asia and Russia, regions with very limited observations; lower uncertainty is associated with regions with a dense station network, such as North America and Europe.

Due to the limitations of spatial kriging in a sparsely monitored region, the observations are often interpolated across very great distances, such as in South America, Africa and Central Asia. This method is not ideal, and instead, information from models can be used to fill in the blanks.

The ozone metric for each model was calculated for each individual grid cell in each year, then averaged over 2008-2014 (except for NASA G5NR-Chem, which was already in fine resolution, but only available for 1 year). Fig 4(a) shows the surface

ozone metric which results from the simple ensemble average of the six models. It was generated from bilinear interpolation of the ozone metric on the standard output grid, by calculating the same metric for each grid cell in each year, averaging over 2008-2014, and then averaging over the 6 models. We refer to this product as the "multi-model mean", and we use it to validate our final product, which should outperform not only each individual model, but also the multi-model mean.

Averaging all 6 models captures the large scale variations of the ozone distribution, however, it misses the most extreme

ozone values that are clearly seen from the observations in the TOAR database. A simple approach to address the uncertainty in the multi-model mean is to calculate the standard deviation for each grid cell from the different models, as shown in Fig 4(b). Higher model uncertainties across South Africa and the Middle East match the pattern of the interpolation uncertainty in Fig. 3(b), and lower model uncertainties occur in regions with dense station networks. These findings suggest that the multi-model mean uncertainty can also reflect the current limited understanding of surface ozone in regions with limited or no observations.

It should be noted that the spatially interpolated observations are smoother in regions with fewer sites, and reveal a more detailed structure in regions with a dense station network. In contrast the multi-model mean is more noisy. Even though we average across multiple years and multiple models, the resulting ozone metric can still be noisy because it is calculated at each grid cell independently. In order to maximize the skill of each model, we restrict the model evaluation to the regional scale in the next section.





## 3.2 Regional model evaluation and Multi-model Composite

To evaluate the performance of each model in a given region, we calculate the mean differences over all grid cells within the region and summarize them with the RMSE. Let $\hat{y}(s_g)$ be the spatially interpolated observations, and $\{\eta_k(s_g); k = 1, \ldots, 6\}$ be the output corresponding to the six ensemble models considered in this paper, then the (normalized) RMSE is given by

$$\text{RMSE}_{rk} = \sqrt{\frac{\sum_{s_g \in Region\ r} \left(\eta_k(s_g) - \hat{y}(s_g)\right)^2}{n}},$$

where $n$ is the number of grid cells in a given region $r$. The first part of Table 2 shows the RMSE statistics for each model by region. The reliability of such an evaluation is limited by the station density in a given region, with greater reliability in a dense network (e.g. USA) and less reliability in a sparse network (e.g. Africa, South America or Australia). On average, GEOSCCM, GFDL-AM3 and G5NR-Chem have the lowest biases in multiple regions; MRI-ESM1r1 also shows low mean biases in certain regions, such as North America and Europe. However, larger model biases can be found in Africa and East Asia.

We next select three regions with extensive monitoring: North America, Europe and East Asia. Fig 5 shows the differences between the spatially interpolated observations and model output in North America. A consistent underestimation can be found in Mexico City for all models. A mild underestimation is also found across much of the USA except for the east coast, and a mixed direction of biases is observed in Canada.

In Europe (Fig 6), the models show mild levels of underestimation across almost the whole region, except for South Europe. Those underestimations are strongest in central Europe. Since Europe has one of the most developed station networks in the world, our spatially interpolated observations have a higher accuracy in terms of spatial distribution, and we can confirm that all models are very similar to the spatially interpolated observations, with low RMSEs and little discrepancy shown in the difference plots.

In East Asia (Fig 7), a major model bias is observed in the Beijing area. However, the amplitudes of biases are smaller for GFDL-AM3 and G5NR-Chem; all models also show a similar bias pattern in Southeast Asia. However, since the observations are relatively sparse in mainland China, the large scale of these estimated biases might be an interpolation artefact.

We argue that the credibility of the model is not entirely decided by the RMSE (i.e. the mean difference): the smoother the difference plots, the easier it is to carry out the model bias correction. Indeed, the observations and model output are not expected to match point by point. We should also expect the model to capture the general pattern of the spatial distribution, rather than point-wise agreement.

The estimated weights from the constrained least squares (Eq 1) are given in the second part of Table 2. Due to fixed underlying spatial structures, this approach tends to give greater weight to a single model (i.e. $\geq 50\%$), the one which provides the best match between its spatial structure and the observation field (e.g., GFDL-AM3 in Africa). Note that this approach disfavors noisy spatial structure, therefore the algorithm gives low weights to MOCAGE, for several reasons. First, the MOCAGE ozone field has not been smoothed by interpolation since it is already produced on the MOCAGE model grid, whereas all other models are interpolated. Secondly, MOCAGE uses a more complete tropospheric chemical scheme with a larger range of species (77 tropospheric species) and has generally a higher reactivity compared to most CCMs (Voulgarakis et al., 2013).



Thus, it tends to provide more temporal and spatial variability. Note that our optimization algorithm estimates the weights according to the similarity of the spatial structures between the interpolated surface and each model. In regions with sparse monitoring the kriged surface can be greatly affected by a few scattered stations, therefore we cannot use the resulting weights to acknowledge the actual model performance in these regions.

The last column of Table 2 shows the averaged and combined RMSEs from the equal weights and the constrained weights. A reduced overall bias can be generally achieved from the constrained weights. This approach suggests that even if a model has a large mean error (e.g. CHASER), it can still be a good simulation if it produces a spatial pattern and curvature similar to the observation field. The offset term $\alpha$ in the optimization Eq 1 is aimed to adjust the overall residuals between the observation field and the multi-model composite into zero mean in each region (regardless of the spatial pattern), therefore if two spatial

fields share a great similarity in terms of their spatial curvatures, but the overall means are different (see the illustration in Fig 1), this term can fill the gap of the overall mean difference between the two fields. On the other hand, if we do not include $\alpha$ in the equation, CHASER will have a smaller weight in the optimization, and GEOSCCM, GFDL-AM3 and G5NR-Chem will dominate most of these regions (not shown).

We combine all models according to the optimum weights from each region for each model. Fig 8(a) shows a map of the

multi-model composite, a weighted blend of the 6 models, with the weighting calculated separately for each continent. Models with greater simulation skill receive higher weighting. The result appears to be a small scale adjustment to the ensemble mean in Fig 4(a). However, when compared to the TOAR observations, the multi-model composite still has clear regional biases.

### 3.3   Bias correction

The last step of producing the final fused surface ozone product is to apply a bias correction to our multi-model composite.

Ideally we would like to apply a bias correction according to raw observations, but most stations are not exactly located on the model grid coordinates. Therefore, to carry out a statistical bias correction on a particular grid, we need to consider the number of nearby stations and the distance to each station. All these considerations aim to deduce a single correction value on a single grid, and thus we are still faced with implementing statistical interpolation. To avoid adding another level of complexity, we decided to set the final fused product to be exactly equal to the spatially interpolated ozone field within 2 degrees of an

observation, as the spatially interpolated ozone field has already accounted for all observations. Due to the global sparseness of observations about 85% of model grid cells over land were not affected by this bias correction. After bias correcting the multi-model composite grid cells within 2-degrees of a TOAR observation site, an immediate benefit is seen for the USA, Mexico City, Italy and East Asia (see Fig 8(b)).

The choice of the correction range, in this case 2 degrees, is a post hoc decision; we also present results with different

correction ranges in supplementary Fig S-3 and S-4. When the radius of influence of the TOAR observations is increased to 5 or more degrees the greatest impact is seen for the Mexico City region, Kenya (the only site is at 3.7 km on the slope of Mt. Kenya) and eastern China. Increasing the radius to 5 or larger degrees does not improve upon the RMSE associated with 2 degrees. Therefore accepting the 2-degree bias correction over other ranges is subjective. An increase of correction range is not ideal because it extrapolates the Mexico City ozone values into the less populated regions of Mexico. It also produces high





ozone in East Africa where no model indicates such an enhancement, likely biased by the high elevation observations on Mt. Kenya.

## 3.4 Validation of the results

Since the raw observations are the only reliable source for validating our results, we align each model grid to observed locations for evaluating the predictive performance. The RMSE of the residuals from all observations in 2008-2014 are displayed in table 3. Note that since the global network of monitoring stations is heavily weighted by North America, Europe, South Korea and Japan, these numbers are not representative of the sparsely monitored regions. We compare the fused surface ozone results to the simple multi-model mean from all 6 models (MMM). Our interim product, i.e. the multi-model composite, is also compared in the table.

Our multi-model composite outperforms the multi-model mean in terms of lowest mean predicted error. Based on the spatially interpolated observations, the resulting multi-model composite takes advantage of the strengths of each model, and achieves a better accuracy. This result proves that our approach is effective, since our interim product has already improved upon the simple multi-model mean. The bias correction further reduces the residuals: this is expected because the spatial kriging algorithm is designed to minimize the difference to observations, thus it has the lowest RMSE (this value is the same for both kriging result and the fused product since we apply the correction based on observed locations). The RMSE of approximately 5 ppb may represent the interannually varying meteorological influence during the years 2008-2014. If this is the case, then 5 ppb may approximate the minimal RMSE that can be achieved in a multi-year analysis.

In summary, the simple multi-model mean method may perform fairly well at the continental or regional scale, but does not provide an accurate representation of the sub-regional structure, this is of course a limitation on the use of coarse resolutions. The weighting applied during the construction of the multi-model composite improved the accuracy but the effect is rather limited, because many small-scale processes are not (yet) resolved by the models. To alleviate the discrepancy further, a statistical method based on local observations is applied to correct the bias. The advantage of our fused surface ozone product over the simple multi-model mean can be clearly seen in Figure 9.

## 4 Discussion

In this article we present a flexible framework to incorporate observations and multiple models for providing an improved estimate of the global surface ozone distribution. Combining multivariate spatial fields in the estimation of ozone distribution is an extension of both the conventional multi-model ensemble approach (i.e. simple average) and a statistical bias correction approach, and was found to improve the prediction of the surface ozone. In summary our approach has the following properties:

1. The multi-year average enables us to reduce the meteorological influence on surface ozone. An extension of this method to time-resolved multi-annual fields can be expected to capture the interannual variability (Shaddick and Zidek, 2015), however such an endeavor would be highly computationally demanding in such a fine resolution setting.





3. Regional model evaluation facilitates a feature selection for multiple competing atmospheric models.

4. Bias correction of the multi-model composite only at a limited range of grid cells avoids using the spatially interpolated ozone field in regions associated with higher levels of uncertainty.

5. For the regions with dense monitoring networks (such as North American, Europe, South Korea and Japan), the final fused product was obtained mainly from the interpolation of observations; elsewhere the final product relied on the multi-model composite through an optimized weight from each model.

Regarding future improvements two key developments can be expected to yield a better estimation of the global surface ozone distribution: Firstly, we can include more simulators for increased leverage. Another way to increase the estimation accuracy is to expand ozone monitoring networks across sparsely sampled regions (Sofen et al., 2016; Schultz et al., 2017; Weatherhead et al., 2017).

Finally, human health studies typically adopt a fine grid resolution, such as a $0.1° \times 0.1°$ grid product, for matching to the gridded world population database. Even though the spatial kriging surrogate can produce the predicted value at any resolution, the accuracy of the fused surface ozone product is still limited by the density of observations around that point, and by the resolution of the global model output.

*Code and data availability.* The sources of the TOAR data and the output from 4 CCMI models are listed in Section 2.1; Requests for output from GFDL-AM3 should be directed to the PIs; G5NR-Chem model outputs are available for download at https://portal.nccs.nasa. gov/datashare/G5NR-Chem/Heracles/12.5km/DATA or can be accessed through the OpenDAP framework at the portal https://opendap.nccs. nasa.gov/dods/OSSE/G5NR-Chem/Heracles/12.5km; The relevant code can be found in R packages for statistical interpolation (R-INLA, Lindgren and Rue (2015)), quadratic programming (limSolve) and spline smoothing (mgcv, Wood (2017)).

## Appendix A: Spatial modeling using the INLA-SPDE approach

In this paper the aim of spatial interpolation is to use (discretized) monitoring observations to build a statistical surrogate model for estimating the ozone distribution over the whole domain on a sphere. We assume that this ozone distribution follows a Gaussian process (GP). A GP is a collection of random variables such that any subset of the observations has a joint Gaussian distribution. It has been widely used in many applications as a machine learning algorithm (Rasmussen and Williams, 2006). In this section we briefly introduce the GP model with a focus on spatial kriging. The GP is a popular choice in spatial statistics because it allows modeling of fairly complicated functional forms, and it also provides a prediction and associated uncertainty at any new location. A common limitation of this interpolation is that the resulting distribution of estimated uncertainty will be lower around individual stations or within dense monitoring networks, and higher in sparsely monitored regions.



Let $Y$ denote an $n$-vector of ozone observations measured at monitoring sites $\mathbf{s}$, then a statistical model for the spatial field can be expressed as: $Y = f(\mathbf{s}) + \varepsilon$, i.e. the model comprises a smooth GP spatial process $f(\mathbf{s})$, capturing spatial association, and an independent normal error $\varepsilon$, which follows a normal error $N(0, \sigma^2)$. This error term can accommodate potential measurement error; on the other hand, kriging without measurement error is usually used for the surrogate of a deterministic model

(i.e. the same input always produces the same output), also known as an emulator (e.g. Conti and O'Hagan (2010)).

The specification of a GP is through its mean function and covariance function, denoting by $f(\mathbf{s}) \sim GP(m(\mathbf{s}), c(\mathbf{s}, \mathbf{s}'))$. To reduce computational intensity, the mean function can be assumed to be a constant $m(\mathbf{s}) = \mu$, thus the resulting spatial distribution is completely defined by the covariance function. A covariance function characterizes correlations between different locations in the spatial process, it is the crucial component in a GP, as it represents our assumptions about the latent field from

which we wish to build a surrogate. Specifically, we use the Matérn covariance function, which is a flexible covariance structure and widely used in spatial statistics (Hoeting et al., 2006; Jun and Stein, 2007, 2008). With the shape parameter $\nu > 0$, the scale parameter $\kappa > 0$, and the marginal precision $\tau^2 > 0$, the covariance structure can be written as:

$$c(\mathbf{h}) = \frac{2^{1-\nu}}{4\pi\kappa^{2\nu}\tau^2\Gamma(\nu+1)}(\kappa\|\mathbf{h}\|)^\nu K_\nu(\kappa\|\mathbf{h}\|), \mathbf{h} \in \mathbb{S}^2,$$

where $\mathbf{h}$ denotes the distance between any two locations: $\mathbf{h} = \mathbf{s} - \mathbf{s}'$, $\Gamma$ is a gamma function, and $K_\nu$ is the modified Bessel

function of the second kind of order $\nu > 0$. The scale parameter $\kappa$ controls the rate of decay of the correlation between two locations as distance increases. Smaller values of $\kappa$, allow for longer ranges over which two sites can be correlated. The smoothness parameter $\nu$ can be seen as the determining behavior of the autocorrelation for observations that are separated by a small distance.

The major disadvantage of using a GP is the computational complexity, which typically involves a cubic complexity in the

number of data points, usually denoted as $O(n^3)$. Several attempts have been made to reduce the computational burden: e.g. Banerjee et al. (2008), Cressie and Johannesson (2008), Rue et al. (2009), Gramacy and Apley (2015). Lindgren et al. (2011) introduced a popular approach in which the Matérn covariance can be approximated by the solution of certain stochastic partial differential equations (SPDE). According to Lindgren et al. (2011), a GP process $f(\mathbf{s})$ with Matérn covariance on a sphere is the solution of the following stationary SPDE:

$$(\kappa^2 - \Delta)^{(\nu+1)/2}\tau f(\mathbf{s}) = \mathcal{W}(\mathbf{s}),$$

where $\Delta$ is the Laplace operator and $\mathcal{W}$ is the Gaussian white noise. The core implication of this mathematical relationship is that an efficient algorithm for solving this SPDE can be applied to approximate the GP (Lindgren et al., 2011).

This INLA-SPDE technique also enables us to quantify the level of nonstationarity in a spatial field by employing basis function representations for both $\kappa$ and $\tau$ (i.e. these quantities are constants in a stationary field). To obtain basic identifiability,

$\kappa(\mathbf{s})$ and $\tau(\mathbf{s})$ are taken to be positive, and their logarithm can be represented as:

$$\log\kappa(\mathbf{s}) = \sum_{k=1}^{p}\theta_k^\kappa\psi_k(\mathbf{s}) \quad \text{and} \quad \log\tau(\mathbf{s}) = \sum_{k=1}^{p}\theta_k^\tau\psi_k(\mathbf{s}), \tag{A1}$$



where $\{\psi_k(\mathbf{s})\}$ is a set of spherical harmonics. The coefficients $\{\theta_k^\kappa\}$ and $\{\theta_k^\tau\}$ represent local variances and correlation ranges (Bolin and Lindgren, 2011; Lindgren et al., 2011). A larger number of basis functions permits the representation of smaller local features.

We now illustrate a series of statistical model fits to select the best predictive ability of the SPDE model. To choose the maximum number of basis functions for the parameters $\kappa$ and $\tau$ in equation A1, model selection techniques must be used. We perform the model selection based on the following criteria:

- RMSE (root-mean-square error): measure of the overall mean difference between predicted values and the observed values;

- DIC (deviance information criterion): the DIC is a measure to compare performance of statistical models by using a criterion based on a trade-off between the goodness of fit and the corresponding complexity of the model. Smaller values of the DIC indicate a better balance between complexity and a good fit;

- GCV (generalized cross validation): the mean residuals in a leave-one-out test. The model that minimizes the average predicted residuals over all the data is selected as the best model.

We estimate 9 statistical models with different numbers of basis functions, presented in Table A1. The simplest model is a stationary Matérn model (we use basis number 0 to represent the $\kappa$ and $\tau$ as constants). The best fit of all criteria occurs when the orders of the basis functions are increased from four to five. We therefore conclude that a model with five spatially varying basis functions is most appropriate for the TOAR observations.

*Competing interests.* The authors have no competing interests to declare.

*Acknowledgements.* This work was funded by the NASA Health and Air Quality Applied Sciences Team (grant #NNX16AQ80G).



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



**Table 1.** List of the ensemble members used in this paper.

| Model ID | Group | Resolution | Meteorological Forcing[†] | References |
|---|---|---|---|---|
| CHASER (MIROC-ESM) | Nagoya University; Japan Agency for Marine-Earth Science and Technology (JAMSTEC), Japan | $2.8° \times 2.8°$ | C2 | Sudo et al. (2002a, b); Watanabe et al. (2011) |
| GEOSCCM | NASA Goddard Space Flight Center, USA | $2.5° \times 2°$ | C2 | Oman et al. (2011) |
| GFDL-AM3 | NOAA Geophysical Fluid Dynamics Laboratory, USA | $2° \times 2°$ | C1SD | Lin et al. (2012, 2014, 2017) |
| G5NR-Chem | NASA Goddard Space Flight Center, USA | $0.125° \times 0.125°$ | ∗ | Hu et al. (2018) |
| MOCAGE | Centre National de Recherches Météorologiques; Météo France, France | $2° \times 2°$ | C2 | Josse et al. (2004); Teyssèdre et al. (2007) |
| MRI-ESM1r1 | Meteorological Research Institute, Japan | $2.8° \times 2.8°$ | C2 | Adachi et al. (2013) |

† Meteorological forcing includes coupled ocean-atmosphere (C2) and nudged to observed reanalysis meteorology (C1SD) in CCMI reference simulations (Morgenstern et al., 2017).

∗ The specification of forcing scenario for this special run is described by Hu et al. (2018).





**Table 2.** RMSEs (averaged errors in a given region) between spatially interpolated observations and each model, along with regionally optimized weights $\{\beta_{rk}:$ for k-th model in region r$\}$ (zero weights are not displayed). Last column shows the RMSEs from equal weighted averages or constrained weights from the multi-model composite. All the numbers are reported in units of ppb (i.e. parts per billion by volume).

| | Regional RMSE | | | | | | Averaged |
| Region | CHASER | GEOSCCM | GFDL-AM3 | G5NR-Chem | MOCAGE | MRI-ESM1r1 | Error |
|---|---|---|---|---|---|---|---|
| Africa | 11.71 | 10.00 | 7.16 | 9.16 | 17.94 | 9.81 | 10.96 |
| N America | 15.48 | 6.13 | 7.10 | 5.80 | 18.57 | 6.59 | 9.95 |
| S America | 15.80 | 15.31 | 8.04 | 9.10 | 17.78 | 8.33 | 12.39 |
| E Asia | 15.43 | 10.81 | 10.87 | 12.80 | 13.44 | 13.17 | 12.75 |
| S/C Asia | 7.53 | 6.98 | 6.17 | 7.04 | 12.10 | 10.89 | 8.45 |
| Europe | 9.22 | 4.83 | 3.52 | 3.25 | 7.52 | 4.84 | 5.53 |
| Oceania | 8.43 | 9.68 | 9.33 | 7.41 | 10.36 | 7.43 | 8.77 |
| Russia | 13.52 | 4.33 | 5.28 | 5.63 | 13.59 | 6.39 | 8.12 |
| | Constrained weights of the multi-model composite | | | | | | Composite |
| Region | CHASER | GEOSCCM | GFDL-AM3 | G5NR-Chem | MOCAGE | MRI-ESM1r1 | Error |
| Africa | 0.26 | 0.01 | 0.70 | - | 0.03 | - | 6.93 |
| N America | 0.32 | - | - | 0.68 | - | - | 5.27 |
| S America | 0.60 | - | - | 0.30 | - | 0.10 | 6.24 |
| E Asia | 0.33 | 0.67 | - | - | - | - | 8.69 |
| S/C Asia | 0.50 | 0.37 | 0.02 | 0.11 | - | - | 4.75 |
| Europe | - | - | 0.52 | 0.48 | - | - | 2.68 |
| Oceania | 0.78 | - | - | 0.22 | - | - | 6.13 |
| Russia | 0.42 | 0.53 | - | - | 0.05 | - | 3.60 |

**Table 3.** RMSE against TOAR observations (i.e. not interpolated ozone) from the multi-model mean (MMM), multi-model composite (from fusion step 2) and the final fused product (from fusion step 3).

| | MMM | Composite | Fusion |
|---|---|---|---|
| E Asia | 11.43 | 8.70 | 5.29 |
| Europe | 7.33 | 5.61 | 4.50 |
| N America | 7.75 | 5.68 | 3.27 |
| Overall∗ | 8.48 | 6.76 | 4.35 |

∗ Overall category includes all available sites around the world.





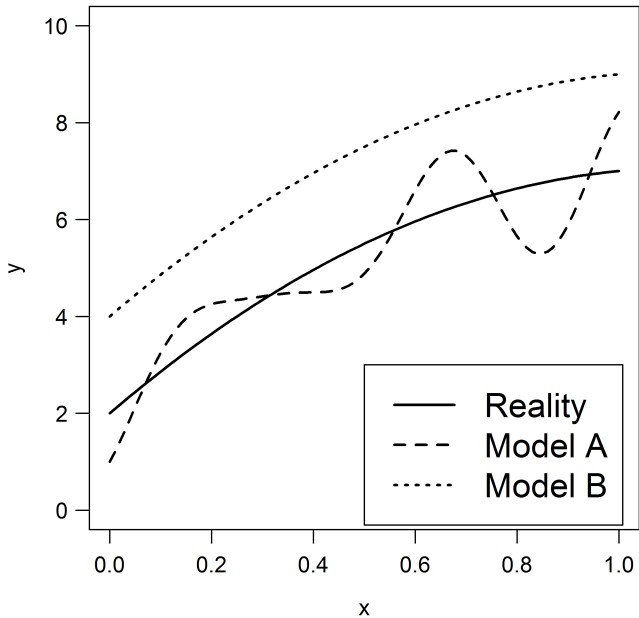

**Figure 1.** Synthetic one dimensional spatial curvatures for the illustration of the models that can be corrected (A) with more effort or (B) with less effort.

**Table A1.** Summary of results from fitting nine candidate statistical models (annual average over 2008-2014).

| # basis | 0 | 1 | 2 | 3 | 4 | 5 | 6 | 7 | 8 |
|---|---|---|---|---|---|---|---|---|---|
| RMSE | 3.82 | 3.17 | 3.18 | 3.23 | 2.90 | 2.52 | 2.76 | 2.76 | 3.44 |
| DIC | -1517 | -1548 | -1556 | -1561 | -1593 | -1621 | -1603 | -1594 | -1565 |
| GCV | 2.78 | 2.64 | 2.62 | 2.60 | 2.50 | 2.43 | 2.44 | 2.48 | 2.60 |



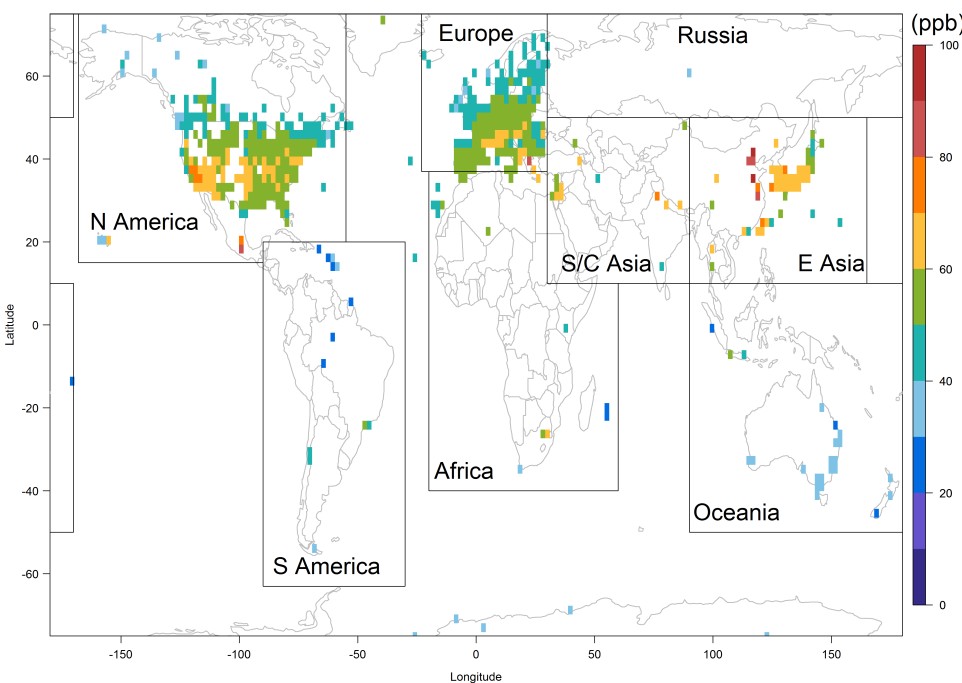

**Figure 2.** TOAR observations where the monitoring locations are discretized to a $2° \times 2°$ grid in 2008-2014.





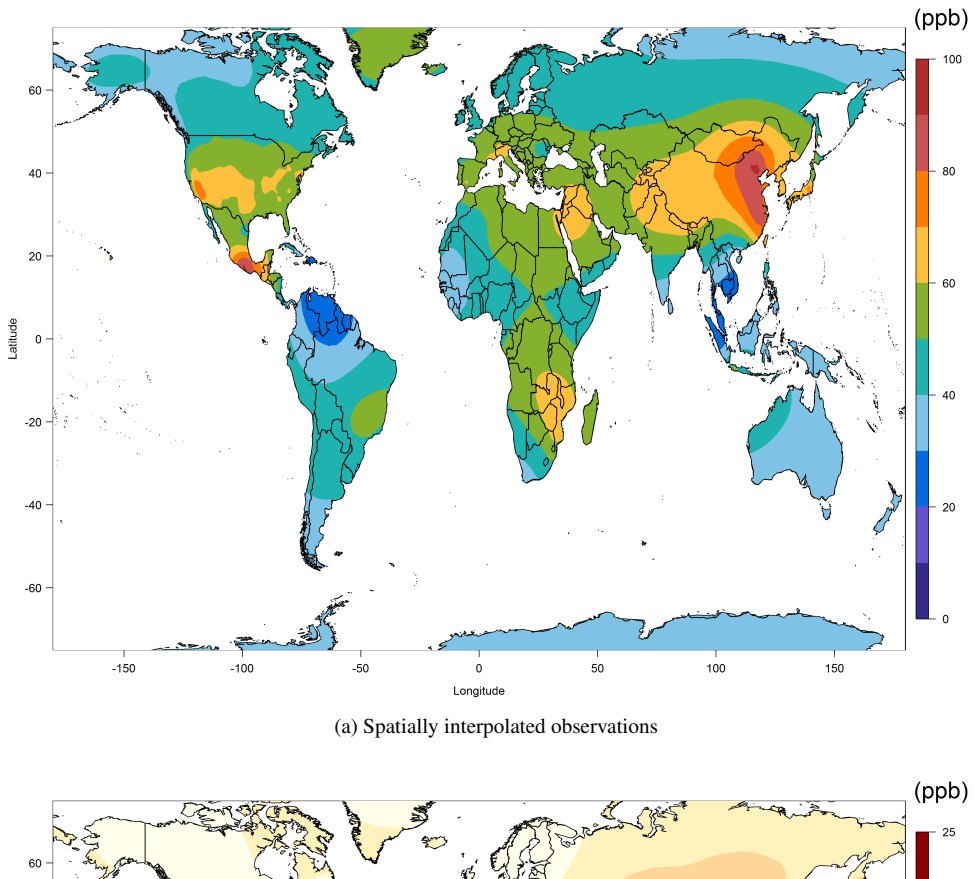

(a) Spatially interpolated observations

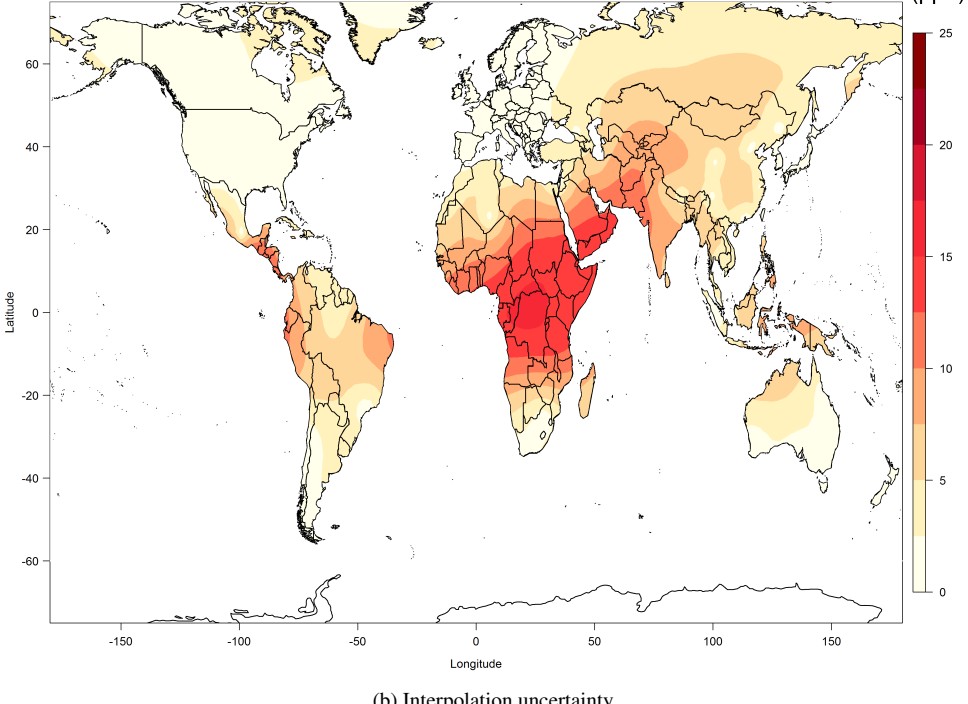

(b) Interpolation uncertainty

**Figure 3.** Estimates of spatially interpolated surface ozone distribution and associated uncertainty (half-width of the 95% credible interval from each cell).



(a) Cell-by-cell mean

(b) Cell-by-cell SD

**Figure 4.** Multi-model mean and standard deviation (SD) in each grid cell from 6 ensemble members.





(a) CHASER

(b) GEOSCCM

(c) GFDL-AM3

(d) G5NR-Chem

(e) MOCAGE

(f) MRIESM1r1

**Figure 5.** Spatial distributions of the ozone metric in North America from each model minus spatially interpolated observations.



(a) CHASER

(b) GEOSCCM

(c) GFDL-AM3

(d) G5NR-Chem

(e) MOCAGE

(f) MRIESM1r1

**Figure 6.** Spatial distributions of the ozone metric in Europe from each model minus spatially interpolated observations.







(a) CHASER

(b) GEOSCCM

(c) GFDL-AM3

(d) G5NR-Chem

(e) MOCAGE

(f) MRIESM1r1

**Figure 7.** Spatial distributions of the ozone metric in East Asia from each model minus spatially interpolated observations.





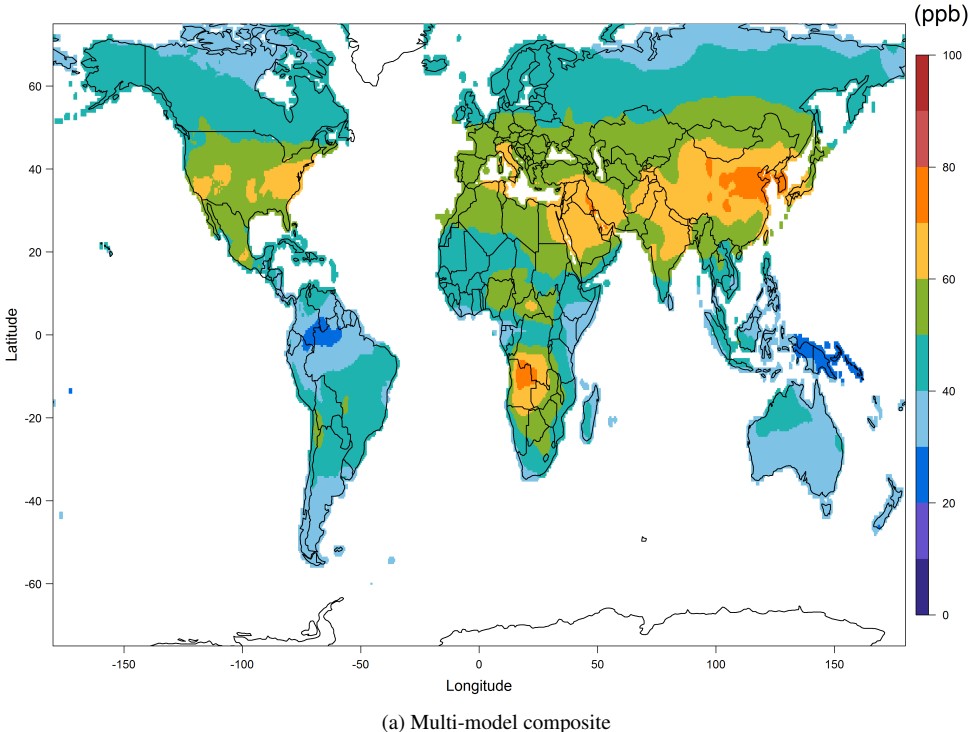

(a) Multi-model composite

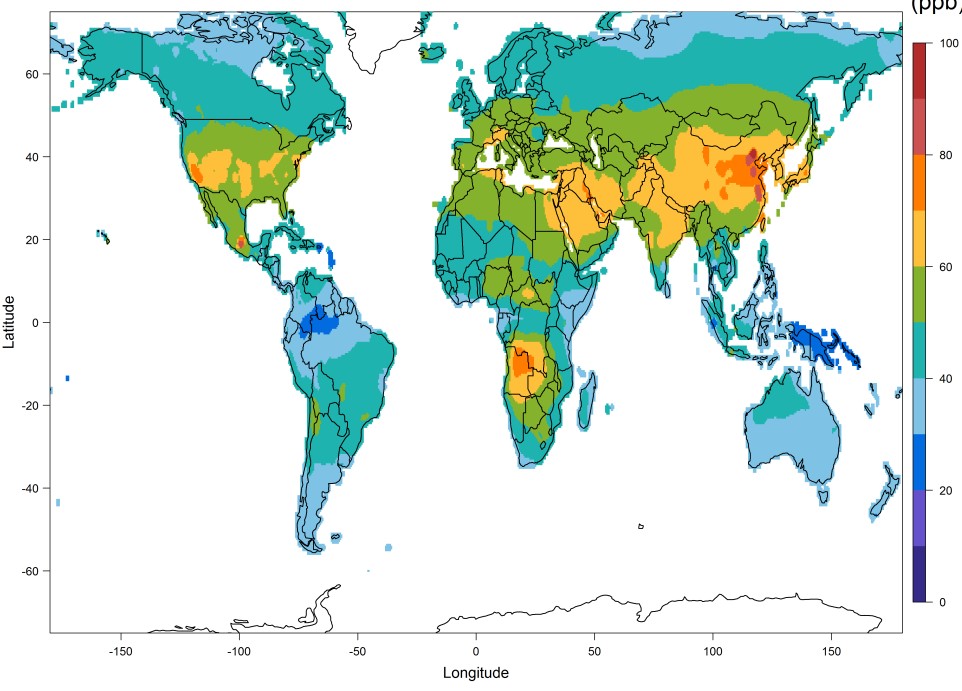

(b) Multi-model composite + bias correction

**Figure 8.** Multi-model composite and bias corrected surface.




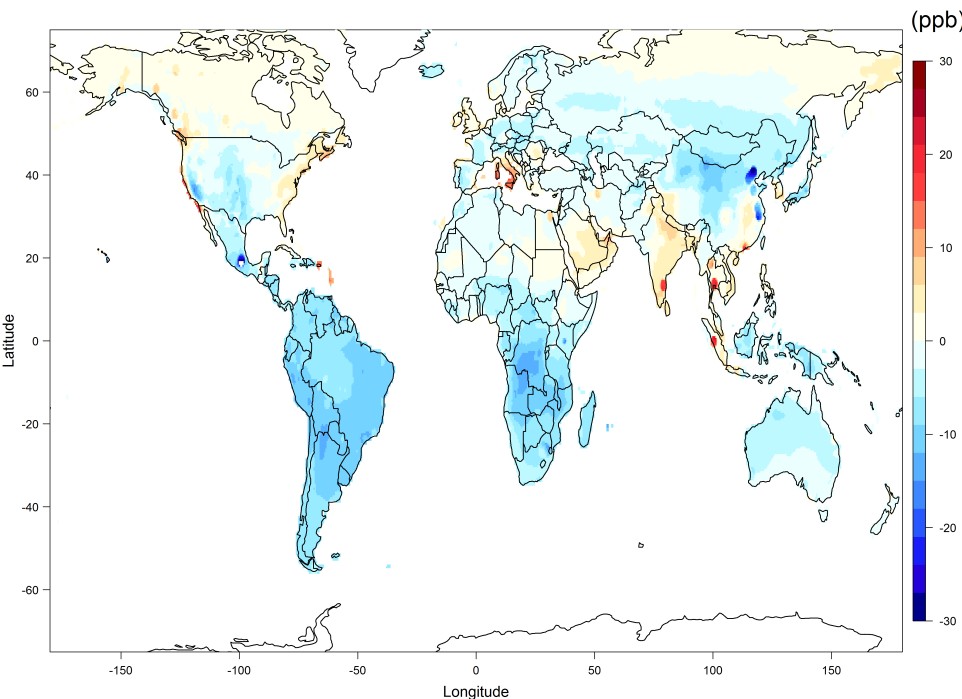

**Figure 9.** Map showing result for multi-model mean minus the fused surface ozone.