# Peer review of "A new method (M3Fusion-v1) for combining observations and multiple model output for an improved estimate of the global surface ozone distribution"

_Geoscientific Model Development, 2018_

## Referee Comment (RC1) · Anonymous Referee #1 · 15 Oct 2018

This manuscript presents a new statistical method for combining observations of surface ozone with model outputs. The manuscript is clearly written and the method is well described. The fused data set represents a significant output that could be useful to analyze the relevance of ozone to health impacts.

The manuscript is nearly ready for publication, but I have several questions and editorial suggestions for the authors, listed below.

1. I suggest to combine Section 2.2 and Section 2.3 into one. Section 2.3 describes the

implementation details but ends up repeating concepts already described in Section 2.2, resulting in poor readability.

2. To create the interpolated field from ozone observations the authors used a Bayesian approach that allows for the quantification of the uncertainty in the gap-filled product.

2.1. Can the authors comment on why they choose not to account for the sampling uncertainty, even though it could be easily estimated from the posterior?

2.2 For example, creating an ensemble of weights (and therefore and ensemble of fused data sets) could be used to explore the impact of poor observational sampling on the fused data set compared to the multi-model mean.

3. In order to compare both the interpolated observations and each models, and the multi-model mean with the fused dataset, I suggest to also plot the empirical variograms, to quantify the differences in the spatial structure.

4. Line 27, page 6: cite the R core development team.

---

## Referee Comment (RC2) · Anonymous Referee #2 · 25 Oct 2018

**General comments. Overall quality**

The article proposes a method for combining measurements from 6 different global models with the aim of generating an improved estimation of the global surface ozone when compared to the estimation obtained by the simple average of these 6 different global models. Hence, this article proposes a method for estimating the weight factor to give to each global model within a weighted average of global models available, and also proposes a method for fusing this result with kriging estimates depending of closeness of locations to monitoring networks. The latter results in an estimated

global surface ozone which is a combination between interpolation-based kriging for areas near monitoring networks, and the proposed weighted average for areas far from monitoring networks. Notwithstanding, results from this article and the final surface for global ozone is estimated by a smoothing splines approach which is applied to the estimation either of the composite model or its fused version. Consequently, the smoothing splines step is the key for the method presented, however it is not explained in the article and authors only dedicate two to three lines to comment about its use to avoid discontinuities in the joints between continental regions.

The authors are trying to address three different problems in geostatistics. First, irregularly spaced sources of information or when the coordinates of the locations from different sources do not match. Second, the lack of information due to observations sparsely distributed or missing locations or almost no information in certain regions. And third, to obtain a better estimate of a surface or compare one estimate with others.

The first problem is more related with interpolation and this is explained well in sections 2.1 and 2.2 of the article.

The second problem is being addressed by the use of global ozone models to obtain a better guess of the non-observed ozone in certain locations. Here the authors propose the composite mean between the global models and its fused version with the interpolation depending on closeness of monitoring network stations which in practice is working as a method for "imputation" of ozone in non-observed locations. The description of the method is mostly well explained (although it is missing important details which I describe in the next section of this report) but the method is not a solution for this problem in areas like Africa or South America where there is not enough information and this is not solved by the composite nor fused method, but by having more measurements. This should be stated clearly in the article. It is difficult to believe that the weight coefficients estimated for Africa or South America would be good estimates given the little sample size available. Nevertheless, the authors could have taken advantage of the dense data available for North America, Europe and East Asia to perform cross-validation and then using the data from South America, Africa and Australia as validation sets of data. This would have provided a rough idea of the quality of the composite estimates to perform the imputation of the ozone level on areas with sparsely distributed observations.

The third problem is poorly or not explained in the article. As presented, the article gives the impression that the main modelling product is the composite mean or its fused version which is confusing since the results are based on a smoothed version and this is not explained in the article. One reader could think that in equation (1) the  $\hat{y}(s_g)$ 's are the "imputed" observations based on the interpolation technique while the composite mean is the proposed model for the ozone level. However, given that later in the article it is expressed that the results are obtained using smoothing splines over the fitted composite mean surface or its fused version, other readers can interpret that actually the fitted composite or fused mean are the imputed observations of the ozone level and the proposed model is the smoothing spline. This is extremely confusing and the article is poorly explained in all this part.

Specific comments. Individual scientific questions/issues

(a) There are important issues which are not addressed by the authors. What is the real role of the smoothing spline applied to the fussed estimation as described in page 8 lines 10-15 and the supplementary material Figure S-2? As the article is presented, this step seems to have a minor role for their proposed method, however it is a key step and the authors did not explain this in detail. In the abstract and along the presentation until section 2.2, the authors' product of this work is a method which relies on a fusion between a weighted average of the 6 global models and the interpolation/kriging step. Nevertheless, from the results it can be deduced that the final product of this work is actually the smoothing spline fit to the surface obtained either by the composite method or its fused version. Therefore, the results presented in Figure 8 (and therefore all related results) rather than being the surface obtained by the multi-model composite and multi-model composite plus bias correction, are respectively the smoothing spline

СЗ

fit to the surface obtained by the multi-model composite and the smoothing spline fit to the surface obtained by the multi-model composite plus bias correction. This must be clearly stated.

(b) What is the interpretation of the parameters in every model discussed? For example:

-What is the practical interpretation of the parameter  $\alpha_r$  in equation (1)? It is related to the general mean over the r-th continental region.

-What is the practical interpretation of the parameter  $\beta_{rk}$  in equation (1)? The cell-bycell average model corresponds to assume  $\beta_{rk} = 1/6$ , thus the same weight is given to each model on each continental region. Then on the composite model  $\beta_{rk}$  can have a meaningful interpretation but the authors do not comment on this.

-What does it mean if  $\beta_{rk} = 0$  (i.e. with respect to the cell-by-cell mean model)? Moreover, what does it mean if  $\beta_{rk} \neq 0$ ? How to interpret if  $\beta_{rk} < 0$ ? How to interpret if  $\beta_{rk} > 0$ ? This is important and connects the cell-by-cell average with the composite mean model proposed. It tells us whether the composite model offers or not a better representation than the average model.

(c) Regarding comment (b), some statistical summaries are not presented in Table 2. For example, what is the significance of each weight/coefficient for the global models and their standard errors or confidence limits? Note that the proposed quadratic programming idea can be seen as a multiple linear regression within each continental region where the  $\hat{y}(s_g)$ 's are seen as the (imputed) observations, the global ozone models  $\eta_{r1}(s_g), \ldots, \eta_{r6}(s_g)$  are seen as predictors or covariates, and the errors are assumed uncorrelated with constant variance. From this approach the authors can obtain variability estimates for the weight coefficients and test their significance.

(d) The authors comment that their composite and fused composite method is better than the simple average (or cell-by-cell average) method. It would be helpful if the

authors presented p-values for a test comparing these two hypotheses.

(e) The authors do not mention the assumption for the mean nor covariance of the smoothing splines model, nor give any details about which type of splines they used (tensor products, thin-plate splines, regression splines, etc.). Did you use penalties? The authors only refer to mgcv R package (Wood, 2017) in line 20 of page 13 and we need to see the code to see what they did, however they should also explain their method, procedure and assumptions in the article. It is the most important modelling they are doing and their results depend on this smoothing splines step.

(f) We can see three different steps in this method. The initial interpolation using INLA, the determination of the weights, and the final smoothing splines using mgcv. INLA and the composite are imputing the ozone measurement on unobserved locations, and the gam function of mgcv package is performing the fit using smoothing splines. In practice, INLA and the smoothing splines are performing the same procedure: interpolation. The only difference is that INLA is based on a triangulation and finite element approach to find a solution. Besides, in both cases the authors are assuming a Matérn covariance function. Therefore, in practice they are fitting an interpolation model to the data (using INLA), and then fitting pretty much the same interpolation model (but using mgcv package) to the previous fit obtained by INLA. Thus, the INLA interpolation is "smoothing" the variation (Figure 3a, page 28), and then an additional smoothing using gam function is being performed (Figure 8a and 8b, page 33, and Figure S-4 in supplementary material). These two fits seem very similar and differences between them can be (visually) attributed mainly to the "variation" generated by the composite mean fit (Figure S-2).

-The first INLA smoothing imputes the ozone at unobserved locations but the resulting "smoothed" process has smaller variation than what we could expect from the original spatial process. Why did the authors not use resampling methods for the imputation step (either before applying the INLA interpolation and/or before applying the smoothing splines fit)? This would have allowed them to keep some spatial variability on the

imputed spatial process, and also evaluate assumptions regarding this spatial variability model.

-Examples of how to implement resampling methods can be found in Liang et al. (2013), and in a more practical presentation by Muñoz et al. (2010). Other approaches based on Expectation-Maximization (EM) algorithm are presented in Schneider (2001).

(g) Regarding the previous comment (f), the authors did not present results about the estimation of the Matérn semivariogram's parameters for the INLA and smoothing splines step. Given that they are performing a "pre-smoothing" of the variation using INLA, it would be expected that the Matérn semivariogram in the smoothing splines step would be modelling a significantly lower amount of spatial variation which might result in almost uncorrelated errors (except in areas where there are peaks or throughs in the process). Is that a good representation or assumption for the global ozone process?

(h) Regarding the modelling part, as presented the article overlooks the real role that the smoothing splines step (page 8 lines 8-15 and Figure S-2 in supplementary material) is playing in the resulting global ozone surface estimated. What is the main modelling technique of their method: the composite method with/without bias correction, or the smoothing splines in question? This is a key issue which cannot be disregarded.

(i) At moments, it is not clear whether the key goal of the article is to propose a novel method to estimate the weights to give to each global ozone model, or to propose an estimated global ozone surface (which is indeed being obtained based on the smoothing splines step). I suggest this is clarified.

(j) The authors mention that the success of their composite mean obtained via the quadratic programming approach depends on the existence of a global ozone model which reproduces correctly the correct curvature on the process (line 30 page 7 – line 5 page 8). This is not necessarily true since equation (1) is only selecting  $\alpha_r$ ,  $\beta_{rk}$  based on a least squares criterion and no regularization conditions on the solution are

being specified, i.e. a curvature penalty. Besides, the curvature of a surface is defined throughout the two-dimensional space in question (the map), and requires the existence of first and second derivatives of the surface. None of these conditions are being established in the article, so that the weights of the composite mean are best only in terms of the "squared distance" between the (imputed) observations and the composite mean at the regular grid of locations being used, not throughout the continuous two-dimensional space.

Additional (tentative) references

Liang, Faming, Yichen Cheng, Qifan Song, Jincheol Park, and Ping Yang. "A resampling-based stochastic approximation method for analysis of large geostatistical data." Journal of the American Statistical Association 108, no. 501 (2013): 325-339.

Muñoz, Breda, Virginia M. Lesser, and Ruben A. Smith. "Applying Multiple Imputation with Geostatistical Models to Account for Item Nonresponse in Environmental Data." Journal of Modern Applied Statistical Methods 9, no. 1 (2010): 27.

Schneider, Tapio. "Analysis of incomplete climate data: Estimation of mean values and covariance matrices and imputation of missing values." Journal of climate 14, no. 5 (2001): 853-871.

---

## Author Comment (AC1) · 30 Jan 2019

We thank both referees for the obvious time and care they put into their reviews, which helped us to revise the manuscript with improved focus and clarity. We have addressed all of the referee comments as described below. In addition, the figures and results were completely revised due to an error that we recently discovered in the particular ozone product that we retrieved from the TOAR Surface Ozone Database and used in this analysis. The product was the monthly mean of the maximum daily 8-hour average (DMA8), calculated for each site in the TOAR database. Close inspection of the product and comparison to daily DMA8 ozone values at individual sites revealed that the sampling of the daily DMA8 values for this particular product was in error, which resulted in monthly means that were biased high. As a result the observed 6-month running mean of the monthly mean DMA8 values used in this analysis was biased high by approximately 25%. The error has been corrected in the TOAR database and in the archived TOAR data products. This analysis has also been updated with the corrected data. However the method for constructing our final fused surface ozone product ($M^3$Fusion) did not change. The final corrected product shows that the atmospheric chemistry models are generally biased high with regards to the 6-month running mean of DMA8. As described in the new concluding paragraph at the end of the manuscript, this is an important result which demonstrates the usefulness of our method for bias correcting model output.

**Anonymous Referee #1**
We thank the reviewer for providing valuable comments on our manuscript. The reviewer comments are shown below in bold font, followed by our response in normal font.

**This manuscript presents a new statistical method for combining observations of surface ozone with model outputs. The manuscript is clearly written and the method is well described. The fused data set represents a significant output that could be useful to analyze the relevance of ozone to health impacts.**
**The manuscript is nearly ready for publication, but I have several questions and editorial suggestions for the authors, listed below.**
**1. I suggest to combine Section 2.2 and Section 2.3 into one. Section 2.3 describes the implementation details but ends up repeating concepts already described in Section 2.2, resulting in poor readability.**
Thanks for the suggestion. We have merged these two sections and removed the overlapping concepts.

**2. To create the interpolated field from ozone observations the authors used a Bayesian approach that allows for the quantification of the uncertainty in the gap-filled product.**
**2.1. Can the authors comment on why they choose not to account for the sampling uncertainty, even though it could be easily estimated from the posterior?**
Accounting for the sampling uncertainty in the data fusion process is a difficult task and according to the referee's comments we now include some discussion of this topic at the end of Section 2:

*"We adopted a regression weighting approach that only accounts for the mean spatial fields of the interpolated ozone and model output, rather than the underlying associated uncertainty. We take this approach due to the prohibitive size of high resolution output  (over 1 million output points for each model), but also due to the lack of a thorough investigation regarding the ideal method for combining models based on different sources of uncertainty. For example, the interpolation uncertainty can be quantified easily through the posterior distribution and considered to be related to measurement error (small scale) or sparse sampling across a region (large scale), however, model uncertainty is a different concept altogether that could result from input uncertainty (e.g. air pollution emissions inventories), or limitations of the transport and chemistry mechanisms within the model (Brynjarsdottir and O'Hagan, 2014). The current interest of this study focuses on a better estimate of mean ozone exposure. Explicit quantification of different sources of model uncertainty and incorporation of this information into the data fusion process presents another level of complexity that cannot be tackled until model uncertainties are better characterized.  Young et al. 2018 provide a current overview of chemistry-climate modelling and discuss the challenges of improving models in light of so many uncertainties."*

**2.2 For example, creating an ensemble of weights (and therefore and ensemble of fused data sets) could be used to explore the impact of poor observational sampling on the fused data set compared to the multi-model mean.**
We expanded the discussion on the differences between our fused product (also model weighting product) and the multi-model mean at the end of Section 3:

*"When interpreting the fused product the reader should consider the following: (1) For a region with an extensive monitoring network, such as the USA, a detailed bias correction can be achieved. We can utilize the observations to accurately reflect many local features (i.e., sub-grid variations) as shown in the ozone pollution hot-spots of southern California and Mexico City. However it should be noted that this improvement is due to bias correction, instead of model weighting; (2) For regions with large observational gaps, such as South America, Africa or Russia, the spatial difference between the fused product and the multi-model mean is rather featureless, because the model weighting can only adjust the overall regional mean according to a few monitoring sites, and cannot address the local variations. Filling large data gaps with the intermediate multi-model composite can indeed avoid the influence of preferential sampling (Diggle et al., 2010; Shaddick and Zidek, 2014), but it is still subject to a high uncertainty due to lack of data."*

**3. In order to compare both the interpolated observations and each models, and the multi-model mean with the fused dataset, I suggest to also plot the empirical variograms, to quantify the differences in the spatial structure.**
Thanks, the variogram is indeed a useful tool to summarize the spatial structure. We added a discussion about variogram in the end of Section 3.3

*"The fused product can be evaluated in terms of spatial correlation using the variogram which assumes that spatial correlation is not a function of absolute location, but only a function of distance (i.e., stationarity). Since spatial variability and continuity from the models are the result of geophysical processes represented by mathematical equations, the variogram must be customized for each field. In addition, the extremely large size of the model output prohibits us from carrying out a standard empirical variogram analysis, which requires calculating the variance of the difference between all pair-wise grid cells.*

*Nevertheless, we provide examples of omnidirectional variograms for the spatial field in North America from each model and product in supplementary Fig. S-5. The standard variogram analysis focuses on the following three parameters: (1) the nugget (variance at zero distance, which represents a sub-grid variation), which is similar for all cases; (2) the sill (total variance of a field), where the variogram value reaches a maximum and levels off The result is very similar for G5NR-Chem, GEOSCCM and GFDL-AM3, while CHASR and MRI-ESM show a larger variance in the spatial field. The reason is that the latter two models produce low ozone in the high latitude region over Canada (see supplementary Fig. S-1), but the former three models simulate relatively higher ozone in the same region, and this difference is reflected by the total variance; (3) the range (a distance where the sill is reached, and beyond that there is no longer spatial correlation): the variogram peak is about 35-40 degrees for the models. Note that a continuously increasing variogram indicates the evidence of non-stationarity in the field, which is the case for SPDE, an issue that we have accounted for. Even though North America has one of the most extensive monitoring networks in the world, some of the remote areas (mostly in Canada) are mainly described by the model output in the final fused product. Therefore the variogram of the fused product is likely adjusted toward the remote areas of Canada as simulated by G5NR-Chem, which provided the largest weighting in North America)."*

**4. Line 27, page 6: cite the R core development team.**
A citation was added to the code and data availability section.

**Anonymous Referee #2**
We thank the reviewer for valuable comments on our manuscript. The comments from the reviewers are below in bold font and we make a response accordingly.

**General comments. Overall quality**
**The article proposes a method for combining measurements from 6 different global models with the aim of generating an improved estimation of the global surface ozone when compared to the estimation obtained by the simple average of these 6 different global models. Hence, this article proposes a method for estimating the weight factor to give to each global model within a weighted average of global models available, and also proposes a method for fusing this result with kriging estimates depending of closeness of locations to monitoring networks. The latter results in an estimated global surface ozone which is a combination between interpolation-based kriging for areas near monitoring networks, and the proposed weighted average for areas far from monitoring networks. Notwithstanding, results from this article and the final surface for global ozone is estimated by a smoothing splines approach which is applied to the estimation either of the composite model or its fused version. Consequently, the smoothing splines step is the key for the method presented, however it is not explained in the article and authors only dedicate two to three lines to comment about its use to avoid discontinuities in the joints between continental regions.**
Thank you for pointing out this issue. We indeed need to clarify that the use of the smoothing splines is not a key method for producing the final fused product, and is only a minor step that we employ to smooth the transition between three regions with sharp discontinuities. This smoothing is only conducted over a 5-degree distance along the boundaries between 2 regions at the 3 locations in the world with the largest discontinuities (see below Figure 1), leaving the rest of the regions unaffected.  This procedure is now more clearly described in Section 2.2 (step 2):

*"This smoothing is carried out using a low rank Gaussian process by the default penalized least square from the function ``gam'' in the R package mgcv (Kammann and Wand, 2003; Wood, 2017), following the examples of Wood and Augustin (2002).The purpose is to merely avoid a sharp and unrealistic (geometric) transition between three regions and to efficiently smooth out the discontinuity, performed in a regular spaced grid only around the geometric boundary. Any region away from the geometric boundary will not be affected by this smoothing, which should be considered as a blending of multiple models without any attempt of bias correction (see supplementary Fig. S-2)."*

We added Figure 1 to the Supplement (Fig. S-2) to illustrate the smoothing procedure, using an expanded color scale to highlight the impact of the spline smoothing on the regional discontinuities.  We only apply the smoothing to 3 regions: one horizontal discontinuity between Russia and East/South Asia, one vertical discontinuity between East and South Asia, and one vertical discontinuity between South Asia and Africa. The discontinuities along the rest of the boundaries between regions were minor and therefore we do not make any further adjustment.

The spatial structure produced from the weighting is not supposed to be create a discontinuity, because the output is smooth. However, a straight line discontinuity is an artifact from our regionalized module in which the models are evaluated for separate regions of the world; the spline smoothing corrects this artifact.

[Figure]

Figure 1: Strong ozone discontinuities, or artifacts, were present along the boundaries between world regions, especially in western China, before a spline smoothing was employed.

**The authors are trying to address three different problems in geostatistics. First, irregularly spaced sources of information or when the coordinates of the locations from different sources do not match. Second, the lack of information due to observations sparsely distributed or missing locations or almost no information in certain regions. And third, to obtain a better estimate of a surface or compare one estimate with others.**

**The first problem is more related with interpolation and this is explained well in sections 2.1 and 2.2 of the article. The second problem is being addressed by the use of global ozone models to obtain a better guess of the non-observed ozone in certain locations. Here the authors propose the composite mean between the global models and its fused version with the interpolation depending on closeness of monitoring network stations which in practice is working as a method for "imputation" of ozone in non-observed locations. The description of the method is mostly well explained (although it is missing important details which I describe in the next section of this report) but the method is not a solution for this problem in areas like Africa or South America where there is not enough information and this is not solved by the composite nor fused method, but by having more measurements. This should be stated clearly in the article. It is difficult to believe that the weight coefficients estimated for Africa or South America would be good estimates given the little sample size available. Nevertheless, the authors could have taken advantage of the dense data available for North America, Europe and East Asia to perform cross-validation and then using the data from South America, Africa and Australia as validation sets of data. This would have provided a rough idea of the quality of the composite estimates to perform the imputation of the ozone level on areas with sparsely distributed observations.**

We modified the text associated with large data gaps in Section 2.2:

*"Above land, large observational gaps are present across Africa, the Middle East, South America, and South and Southeast Asia, where the spatial interpolation is generally too uncertain to yield a reliable surface ozone approximation. The ozone estimates in these regions must come from either models or distant observations, neither of which is ideal to solve this issue. As a compromise strategy we fill these gaps with a weighted model product evaluated by the interpolated ozone observations."*

Following the recommendations of the referee we have expanded this discussion at the end of Section 3:

*"The advantage of our fused surface ozone product over the simple multi-model mean can be clearly seen in Figure 8. When interpreting the fused product the reader should consider the following: (1) For a region with an extensive monitoring network, such as the USA, a detailed bias correction can be achieved. We can utilize the observations to accurately reflect many local features (i.e., sub-grid variations) as shown in the ozone pollution hot-spots of southern California and Mexico City. However it should be noted that this improvement is due to bias correction, instead of model weighting; (2) For regions with large observational gaps, such as*

*South America, Africa or Russia, the spatial difference between the fused product and the multi-model mean is rather featureless, because the model weighting can only adjust the overall regional mean according to a few monitoring sites, and cannot address the local variations. Filling large data gaps with the intermediate multi-model composite can indeed avoid the influence of preferential sampling (Diggle et al., 2010; Shaddick and Zidek, 2014), but it is still subject to a high uncertainty due to lack of data."*

The cross validation technique is indeed a common criterion for assessing the spatial fits, and we used a simple leave one out (LOO) cross validation for assessing our model (the GCV score in table A1). However this score represents an overall LOO error, and doesn't allow for an observation in sparsely sampled region, such as Africa, to receive a lower fitted error than any other observation (and it should not because it would be a conceptual prejudice). We prefer to avoid this type of analysis as we cannot explicitly quantify the representativeness of every single site.

**The third problem is poorly or not explained in the article. As presented, the article gives the impression that the main modelling product is the composite mean or its fused version which is confusing since the results are based on a smoothed version and this is not explained in the article. One reader could think that in equation (1) the yˆ(sg)'s are the "imputed" observations based on the interpolation technique while the composite mean is the proposed model for the ozone level. However, given that later in the article it is expressed that the results are obtained using smoothing splines over the fitted composite mean surface or its fused version, other readers can interpret that actually the fitted composite or fused mean are the imputed observations of the ozone level and the proposed model is the smoothing spline. This is extremely confusing and the article is poorly explained in all this part.**
Thanks for the commentary. We hope we have clarified this concern over the smoothing splines, as described above. The model composite is indeed made from the models, while the smoothing splines only play a minor role for the purpose of removing the straight line discontinuities in Asia and Africa. We further clarified this point in Section 2.2 (step 2):

*"It should also be noted that the INLA-SPDE technique in step 1 is applied to the observations, while the smoothing spline is only applied to the boundaries between regions of the model composite, not directly involving any observations."*

**Specific comments. Individual scientific questions/issues**

**(a) There are important issues which are not addressed by the authors. What is the real role of the smoothing spline applied to the fussed estimation as described in page 8 lines 10-15 and the supplementary material Figure S-2? As the article is presented, this step seems to have a minor role for their proposed method, however it is a key step and the authors did not explain this in detail. In the abstract and along the presentation until**

**section 2.2, the authors' product of this work is a method which relies on a fusion
between a weighted average of the 6 global models and the interpolation/kriging step.
Nevertheless, from the results it can be deduced that the final product of this work is
actually the smoothing spline fit to the surface obtained either by the composite method
or its fused version. Therefore, the results presented in Figure 8 (and therefore all related
results) rather than being the surface obtained by the multi-model composite and
multi-model composite plus bias correction, are respectively the smoothing spline fit to
the surface obtained by the multi-model composite and the smoothing spline fit to
the surface obtained by the multi-model composite plus bias correction. This must be
clearly stated.**

To illustrate the limited impact of the spline smoothing on just three regional boundaries we
have included figure S-2 in the Supplement.

**(b) What is the interpretation of the parameters in every model discussed? For example:
-What is the practical interpretation of the parameter $\alpha_r$ in equation (1)? It is related to the
general mean over the r-th continental region.
-What is the practical interpretation of the parameter $\beta_{rk}$ in equation (1)? The cell-by-cell
average model corresponds to assume $\beta_{rk} = 1/6$, thus the same weight is given to each
model on each continental region. Then on the composite model $\beta_{rk}$ can have a
meaningful interpretation but the authors do not comment on this.
-What does it mean if $\beta_{rk} = 0$ (i.e. with respect to the cell-by-cell mean model)?
Moreover, what does it mean if $\beta_{rk} \ne 0$? How to interpret if $\beta_{rk} < 0$? How to interpret if
$\beta_{rk} > 0$? This is important and connects the cell-by-cell average with the composite mean
model proposed. It tells us whether the composite model offers or not a better
representation than the average model.**

Since the interpolated observations and models use the same ozone metric with the same units,
it makes sense that we restrict the coefficient of the covariate to a range between [0, 1] and
summed to 1:

- From a regression point of view, if we only include beta (weight) without a constant, the
residuals will have a biased mean, so the alpha term will force the overall residuals to have a
mean value of zero.
- Any positive value of beta should be seen as a significant component of the model composite.
- If beta is zero, it means this particular model makes no contribution to the model composite.
The coefficient is not permitted to have a negative value since a negative value doesn't have a
physical meaning.

We modified the interpretation in Section 2.2 (step 2):

*"Note that since the interpolated observations and models use the same ozone metric with the
same units, we thus constrain the weights to be positive and sum to 1 for better physical
interpretability, such that the most accurate models receive the higher weight. A constant offset
$\alpha_r$ is included to guarantee that the residuals from this optimization have a zero mean."*

**(c) Regarding comment (b), some statistical summaries are not presented in Table 2. For example, what is the significance of each weight/coefficient for the global models and their standard errors or confidence limits? Note that the proposed quadratic programming idea can be seen as a multiple linear regression within each continental region where the yˆ(sg)'s are seen as the (imputed) observations, the global ozone models ηr1(sg), . . . , ηr6(sg) are seen as predictors or covariates, and the errors are assumed uncorrelated with constant variance. From this approach the authors can obtain variability estimates for the weight coefficients and test their significance.**

There is no standard or consensus methodology for combining models based on the uncertainty (i.e., standard errors), and we are unable to adjust the weights based on the standard errors. For example, no matter what value of standard error is associated with a 0 coefficient, it will still have a 0 weight, which doesn't allow us to properly interpret the variability. This is why we restrict the coefficients to a range between [0, 1] and summed to 1; this arrangement forces the coefficient of an insignificant predictor to be 0, and any positive coefficient should be seen as a significant contribution to the model composite. We added a discussion on the difficulty of combining models based on the uncertainties to the Section 2.2:

*"We adopted a regression weighting approach that only accounts for the mean spatial fields of the interpolated ozone and model output, rather than the underlying associated uncertainty. We take this approach due to the prohibitive size of high resolution output  (over 1 million output points for each model), but also due to the lack of a thorough investigation regarding the ideal method for  combining models based on different sources of uncertainty. For example, the interpolation uncertainty can be quantified easily through the posterior distribution and considered to be related to measurement error (small scale) or sparse sampling across a region (large scale), however, model uncertainty is a different concept altogether that could result from input uncertainty (e.g. air pollution emissions inventories), or limitations of the transport and chemistry mechanisms within the model (Brynjarsdottir and O'Hagan, 2014). The current interest of this study focuses on a better estimate of mean ozone exposure. Explicit quantification of different sources of model uncertainty and incorporation of this information into the data fusion process presents another level of complexity that cannot be tackled until model uncertainties are better characterized.  Young et al. 2018 provide a current overview of chemistry-climate modelling and discuss the challenges of improving models in light of so many uncertainties."*

**(d) The authors comment that their composite and fused composite method is better than the simple average (or cell-by-cell average) method. It would be helpful if the authors presented p-values for a test comparing these two hypotheses.**

To the best of our knowledge the use of a hypothesis test or p-value for comparing spatial model fits (or climate model performance) is not an ideal approach and not discussed in the literature. This is largely because the kriging procedure (or Gaussian process) is the result of machine learning, so there is no corresponding "hypothesis testing" concept for the Gaussian process.  Rather, computer scientists tune the parameters to yield the best output.

The most common practice to measure the performance of a model is directly comparing the root mean square error (RMSE, as shown in tables 2 and 3) between observations and output, and quantifying the percentage of improvement. The report of the physical quantity, such as RMSE shown in the same unit as the ozone metric, should be more meaningful than potentially misleading p-values.

 **(e) The authors do not mention the assumption for the mean nor covariance of the smoothing splines model, nor give any details about which type of splines they used (tensor products, thin-plate splines, regression splines, etc.). Did you use penalties? The authors only refer to mgcv R package (Wood, 2017) in line 20 of page 13 and we need to see the code to see what they did, however they should also explain their method, procedure and assumptions in the article. It is the most important modelling they are doing and their results depend on this smoothing splines step.**
As described above, the spline smoothing was just a very minor component of our method, only used to smooth 3 geographical discontinuities. We used a particularly simple form of the Matern covariance function suggested by Kammann and Wand (2003), and we added the details to Section 2.2.

The main command to perform this smoothing is given by (a complete code can be found in supplementary material):

```
mod = gam(composite ~ s(lon,lat, bs="gp", k=180), data=sm, method="REML",
na.action='na.omit')
```

Since the removal of a straight line discontinuity is the only concern, any spline model should achieve this goal as long as it can handle the high resolution output. We chose this Matern spline merely because it is simple and efficient for high resolution output.

**(f) We can see three different steps in this method. The initial interpolation using INLA, the determination of the weights, and the final smoothing splines using mgcv. INLA and the composite are imputing the ozone measurement on unobserved locations, and the gam function of mgcv package is performing the fit using smoothing splines. In practice, INLA and the smoothing splines are performing the same procedure: interpolation. The only difference is that INLA is based on a triangulation and finite element approach to find a solution. Besides, in both cases the authors are assuming a Matérn covariance function. Therefore, in practice they are fitting an interpolation model to the data (using INLA), and then fitting pretty much the same interpolation model (but using mgcv package) to the previous fit obtained by INLA. Thus, the INLA interpolation is "smoothing" the variation (Figure 3a, page 28), and then an additional smoothing using gam function is being performed (Figure 8a and 8b, page 33, and Figure S-4 in supplementary material). These two fits seem very similar and differences between them can be (visually) attributed mainly to the "variation" generated by the composite mean fit (Figure S-2).**

Yes the INLA and smoothing spline are indeed performing the same procedure, but as described above the spline smoothing is only used under very limited circumstances to smooth 3 regional boundary discontinuities. We also added a note in Section 2.2 (step 2):

*"It should be noted that the INLA-SPDE technique in step 1 is applied to the observations, while the smoothing spline is only applied to the boundaries between regions of the model composite, not directly involving any observations."*

**-The first INLA smoothing imputes the ozone at unobserved locations but the resulting "smoothed" process has smaller variation than what we could expect from the original spatial process. Why did the authors not use resampling methods for the imputation step (either before applying the INLA interpolation and/or before applying the smoothing splines fit)? This would have allowed them to keep some spatial variability on the imputed spatial process, and also evaluate assumptions regarding this spatial variability model.**
**-Examples of how to implement resampling methods can be found in Liang et al. (2013), and in a more practical presentation by Muñoz et al. (2010). Other approaches based on Expectation-Maximization (EM) algorithm are presented in Schneider (2001).**

When comparing Figures 1 and 2(a), the INLA interpolation can reproduce the hot spots in East China, Mexico City and LA, but it indeed missed the highest ozone in the Beijing Metropolitan area. This result is related to the degree of smoothing that we allow for the spatial field. The success of reproducing ozone in East China, Mexico City and LA is due to these locations having multiple grids with high ozone observations. However, there is only one grid with high ozone observed in Beijing, and half of the grid points around that spot do not observe high ozone, therefore the smoothing of the spatial field missed this single hot spot. Reducing the degree of smoothing in order to capture the ozone hot spot in Beijing Metropolitan would introduce more noise and unrealistic peaks in other regions, and also increase the computational burden.

We choose the INLA-SPDE technique for interpolation merely because it can incorporate the non-stationary component in an easy way (illustrated in the appendix). No matter the details of the INLA-SPDE, resampling method, or other approaches mentioned in Section 2.2 (covariance tapering, low rank, spectral representation, likelihood approximation…), they are all special cases of a more general Gaussian process designed to alleviate the large sample size (n) problem. Parts of these approaches are only proven to be efficient on regional or national scales, and they are not necessarily adequate on the global scale. For example in the differential manifold, a covariance model is positive definite on a plane, it is not guarantee that will also valid in a sphere (Gneiting, 2013).

The idea of resampling is similar to the leave-n-out validation, it keeps iteratively removing a portion of data and re-fitting the model until the error is minimized. So it is an algorithm to improve the fit to the data (we cannot leave the data out without an actual observation).

However, the similar but much simpler leave-1-out validation is used for evaluating the interpolation performance (the GCV score in table A1). We added the citations accordingly in the manuscript, but Munoz et al. (2010) focused on the discussion of type and mechanism of missing data, which is a bit far from our topic.

**(g) Regarding the previous comment (f), the authors did not present results about the estimation of the Matérn semivariogram's parameters for the INLA and smoothing splines step. Given that they are performing a "pre-smoothing" of the variation using INLA, it would be expected that the Matérn semivariogram in the smoothing splines step would be modelling a significantly lower amount of spatial variation which might result in almost uncorrelated errors (except in areas where there are peaks or throughs in the process). Is that a good representation or assumption for the global ozone process?**
The INLA package provides a more general class of model that can specify a spatially varying nugget and sill, which would be more flexible than the variogram approach that assumes a fixed nugget and sill over the whole spatial field. From a series expansion of spherical harmonics in Eq A1, we used several basis functions to select the best statistical model in table A1; for further details please refer to the appendix. We also added a discussion about variograms to the end of Section 3.3

*"The fused product can be evaluated in terms of spatial correlation using the variogram which assumes that spatial correlation is not a function of absolute location, but only a function of distance (i.e., stationarity). Since spatial variability and continuity from the models are the result of geophysical processes represented by mathematical equations, the variogram must be customized for each field. In addition, the extremely large size of the model output prohibits us from carrying out a standard empirical variogram analysis, which requires calculating the variance of the difference between all pair-wise grid cells.*

*Nevertheless, we provide examples of omnidirectional variograms for the spatial field in North America from each model and product in supplementary Fig. S-5. The standard variogram analysis focuses on the following three parameters: (1) the nugget (variance at zero distance, which represents a sub-grid variation), which is similar for all cases; (2) the sill (total variance of a field), where the variogram value reaches a maximum and levels off The result is very similar for G5NR-Chem, GEOSCCM and GFDL-AM3, while CHASR and MRI-ESM show a larger variance in the spatial field. The reason is that the latter two models produce low ozone in the high latitude region over Canada (see supplementary Fig. S-1), but the former three models simulate relatively higher ozone in the same region, and this difference is reflected by the total variance; (3) the range (a distance where the sill is reached, and beyond that there is no longer spatial correlation): the variogram peak is about 35-40 degrees for the models. Note that a continuously increasing variogram indicates the evidence of non-stationarity in the field, which is the case for SPDE, an issue that we have accounted for. Even though North America has one of the most extensive monitoring networks in the world, some of the remote areas (mostly in Canada) are mainly described by the model output in the final fused product. Therefore the*

*variogram of the fused product is likely adjusted toward the remote areas of Canada as simulated by G5NR-Chem, which provided the largest weighting in North America)."*

The multi-model mean and composite (Figures 3(a) and 7(a)) show a lower spatial variation, which is unrelated to INLA or the Matern function, since they are completely made/weighted from the model output. Based on the comparison of the similarity between Figures 1 and 2(a), we see that the interpolation reproduces many local features, a result that we view as successful.

**(h) Regarding the modelling part, as presented the article overlooks the real role that the smoothing splines step (page 8 lines 8-15 and Figure S-2 in supplementary material) is playing in the resulting global ozone surface estimated. What is the main modelling technique of their method: the composite method with/without bias correction, or the smoothing splines in question? This is a key issue which cannot be disregarded.**
As described above, the smoothing splines are only a minor component of the overall analysis. Our final fused product is the result of three important steps, all of which are necessary. We modified the following paragraph in the Introduction that summarizes the key steps in this process:

"This paper presents a new statistical approach (M3Fusion) for combining surface ozone output from multiple atmospheric chemistry models with all available surface ozone observations to produce a global surface ozone distribution with greater accuracy than the multi-model ensemble mean. As described in greater detail below, this fused surface ozone product is constructed in three steps: 1) Ozone observations from all available surface ozone monitoring sites around the world are spatially interpolated to a smooth global field; 2) For each of 8 continental regions of the world 6 global atmospheric chemistry models are evaluated against the interpolated observed ozone field by a quadratic programming optimization, with the most accurate models receiving the highest weight. A locally confined spline interpolation is used at the regional boundaries to avoid unphysical step changes; 3) finally, the global ozone field derived from the polynomial equation is bias corrected, but only within a limited distance from available observations. The final product is based on the annual maximum of the 6-month running mean of the monthly average daily maximum 8-hour average mixing ratios (DMA8), a metric that can be used to estimate human mortality due to long-term ozone exposure (Turner et al., 2016; Malley et al., 2017; Seltzer et al., 201; Shindell et al.,2018)."

**(i) At moments, it is not clear whether the key goal of the article is to propose a novel method to estimate the weights to give to each global ozone model, or to propose an estimated global ozone surface (which is indeed being obtained based on the smoothing splines step). I suggest this is clarified.**
The goal is to create a best estimate surface for the global ozone concentration, and this is done through a three-step process (which we clarified in the paragraph as above), and we also added a note in Conclusion that this method can be used for different applications:

*"The application of our methodology focuses on, but is not limited to, a particular ozone metric relevant for quantifying the impact of long-term ozone exposure on human health. We expect that this framework could also be applied to other ozone metrics relevant to crop production or natural vegetation (Lefohn et al., 2018; Mills et al., 2018), or any other trace gas provided adequate in situ observations are available for model evaluation."*

**(j) The authors mention that the success of their composite mean obtained via the quadratic programming approach depends on the existence of a global ozone model which reproduces correctly the correct curvature on the process (line 30 page 7 – line 5 page 8). This is not necessarily true since equation (1) is only selecting αr, βrk based on a least squares criterion and no regularization conditions on the solution are being specified, i.e. a curvature penalty. Besides, the curvature of a surface is defined throughout the two-dimensional space in question (the map), and requires the existence of first and second derivatives of the surface. None of these conditions are being established in the article, so that the weights of the composite mean are best only in terms of the "squared distance" between the (imputed) observations and the composite mean at the regular grid of locations being used, not throughout the continuous two-dimensional space.**

Thanks for the suggestion, we indeed have not discussed about spatial curvature and first/second derivatives of the surface. We thus removed this paragraph and added a clarification in p7:

*"The weights are optimized in terms of the squared distance between the interpolated ozone and multi-model output. A different criterion of optimization, such as mean absolute error, can be established accordingly."*

**Additional (tentative) references**
**Liang, Faming, Yichen Cheng, Qifan Song, Jincheol Park, and Ping Yang. "A resampling-based stochastic approximation method for analysis of large geostatistical data." Journal of the American Statistical Association 108, no. 501 (2013): 325-339.**
**Muñoz, Breda, Virginia M. Lesser, and Ruben A. Smith. "Applying Multiple Imputation with Geostatistical Models to Account for Item Nonresponse in Environmental Data." Journal of Modern Applied Statistical Methods 9, no. 1 (2010): 27.**

[revised manuscript text omitted]

*Corresponding author: kai-lan.chang@noaa.gov

**List of Figures**

[Figure]

**Figure S-1:** Global distributions of the ozone metric from ensemble members.

[Figure]

**(a)** Before spline smoothing

**(b)** After spline smoothing

**Figure S-2:** Strong ozone discontinuities, or artefacts, were present along the geometric boundaries, especially in western China, before a spline smoothing was employed. The smoothing is only applied to 3 regions: one horizontal discontinuity between Russia and East/South Asia, one vertical discontinuity between East and South Asia, and one vertical discontinuity between South Asia and Africa.

[Figure]

**Figure S-3:** The multi-model bias corrected surface under different ranges of correction radius.

[Figure]

**(a)** 2 degrees

**(b)** 5 degrees

**(c)** 10 degrees

**(d)** 15 degrees

**Figure S-4:** Amplitudes of multi-model bias correction under different ranges of correction radius.

[Figure]

**Figure S-5:** The empirical variogram of ozone metric in North America from each model and product.

---

## Author Comment (AC2) · 30 Jan 2019

Please see the attached pdf file for the response to both reviewers.

Please also note the supplement to this comment:
https://www.geosci-model-dev-discuss.net/gmd-2018-183/gmd-2018-183-AC2-supplement.pdf